# On the use of streamflow transformations for hydrological model calibration

Guillaume Thirel[1], Léonard Santos[1], Olivier Delaigue[1], and Charles Perrin[1]

[1]Université Paris-Saclay, INRAE, UR HYCAR, 92160 Antony, France

**Correspondence:** Guillaume Thirel (guillaume.thirel@inrae.fr)

**Abstract.** The calibration of hydrological models through the use of automatic algorithms aims at identifying parameter sets that minimize the deviation of simulations from observations (often streamflows). Further, the choice of objective function (i.e. the criterion or combination of criteria to optimize) can significantly impact the parameter set values identified as optimal by the algorithm. This article discusses how mathematical transformations, which are sometimes applied to the target variable before calculating the objective function, impact model simulations. Such transformations, for example square root or logarithmic, aim at increasing the weight of errors made in specific ranges of the hydrograph. We show in a catchment set that the impact of these transformations on the obtained time series can sometimes be different from their expected behaviour. Extreme transformations, such as squared or inverse of squared transformations, lead to models that are specialized for extreme streamflows but show poor performance outside the range of the targeted streamflows and are less robust. Other transformations, such as the power 0.2, the Box–Cox and the logarithmic transformations, can be qualified as more generalist and show a good performance for the medium range of streamflows, along with an acceptable performance for extreme streamflows.

## 1 Introduction

Hydrological models are essential tools for hypothesis testing and process understanding (Rosbjerg and Madsen, 2006) but also for very practical applications such as flood or low-flow forecasting, water resources management or the assessment of climate change impacts. Despite the long-lasting efforts of hydrologists, there is consensus in the community that no universal hydrological model structure exists and it is doubtful whether it will ever be found. This has motivated a proliferation of flexible modelling platforms such as FUSE (Clark et al., 2008), SUPERFLEX (Fenicia et al., 2011), Noah-MP (Niu et al., 2011), SUMMA (Clark et al., 2015b, a, 2021b), MARRMoT (Knoben et al., 2019), Raven (Craig et al., 2020) or airGR (Coron et al., 2017). In order to fit specific applications and due to the wide catchment diversity and the various targeted streamflow ranges, performing a calibration of model parameters is generally necessary. The calibration process usually relies on the use of one or more criteria, i.e. a numerical metric of the model error, which is used as an objective function. The choice of this optimization criterion is subjective (Mendoza et al., 2016; Fowler et al., 2018; Melsen et al., 2019), since it depends on

various aspects (application objective, model characteristics, etc.), and two different criteria will impact the calibration process differently and will lead to different optimal parameter sets and performances (Booij and Krol, 2010). In addition, these criteria suffer from flaws leading to their incorrect use by modellers (Clark et al., 2021a), and each modeller has their own vision of what constitutes a good model or hydrograph and how it translates into a numerical criterion (Crochemore et al., 2015).

While criteria are usually calculated for comparing raw simulated and observed streamflow time series, a wide range of transformations have been introduced in the literature (Bennett et al., 2013), which consist in using a mathematical function in order to transform both simulated and observed time series. These transformations rely on the fact that they distort the observed and simulated time series and their properties in such a way as to expect that the related errors are similarly distorted. This is illustrated in Fig. 1, where in panel a, the larger errors between the observed and simulated time series mostly occur for high-flow periods (pink shaded area), while in panel b, with log-transformed flows, these errors are much larger over low-flow periods (green shaded areas).

Since many metrics rely on squared errors (e.g. root mean square error or Nash–Sutcliffe efficiency, Nash and Sutcliffe, 1970) and therefore are known to emphasize the most important errors (Sorooshian and Dracup, 1980), a large set of transformations were proposed for better representation of low flows. A non-exhaustive list of transformations is listed in Table 1, with the square root, the logarithmic, the reciprocal of squared root, the inverse or other power–law transformations being the most popular. Other studies used the Box–Cox transformation with different $\lambda$ values or combined several of the transformations listed above.

While the choice of transformations is wide and the theoretical basis is sound (as shown in Fig. 1), there is not an extensive literature discussing the merits of the transformation approach. Pushpalatha et al. (2012) justified the use of transformations by several authors through the fact that "the sum of squared residuals calculated on the logarithms of flow values" reduces "the biasing towards peak flows". They investigated which range of streamflows leads to the largest part of errors. Smith et al. (2014) showed that a transformation called "flow-corrected time", designed to provide greater weight to time periods with larger hydrologic flux, results in improved fits, compared to a baseline untransformed case and the logarithmic transformation, over the time periods that dominate hydrologic flux. Peña-Arancibia et al. (2015) showed that a squared root transformation with the Nash–Sutcliffe efficiency leads to higher performance, both on the calibration and on the evaluation period, and a reduced parameter uncertainty compared with no transformation or a logarithmic transformation. Sadegh et al. (2018) investigated the role of several transformations in three catchments and two models and deduced that some data transformations might be more helpful for the evaluation of model performance and the analysis of model behaviour than for calibration.

To the best of our knowledge, only a few studies have assessed thoroughly the use and choice of transformations. For example, Krause et al. (2005) stated that they used the logarithmic transformation on the Nash–Sutcliffe efficiency "to reduce the problem of the squared differences [...]. Through the logarithmic transformation of the runoff values the peaks are flattened and the low flows are kept more or less at the same level. As a result the influence of the low flow values is increased in comparison to the flood peaks". Chiew et al. (1993) used a power $0.2$ transformation and justified it by the fact that "it generally leads to constant variances (values of $SIM^{0.2} - REC^{0.2}$ are similar for all flow volumes) in many of the temperate catchments where models have been applied by the authors". Oudin et al. (2006) reported that "it is common practice in hydrology to use

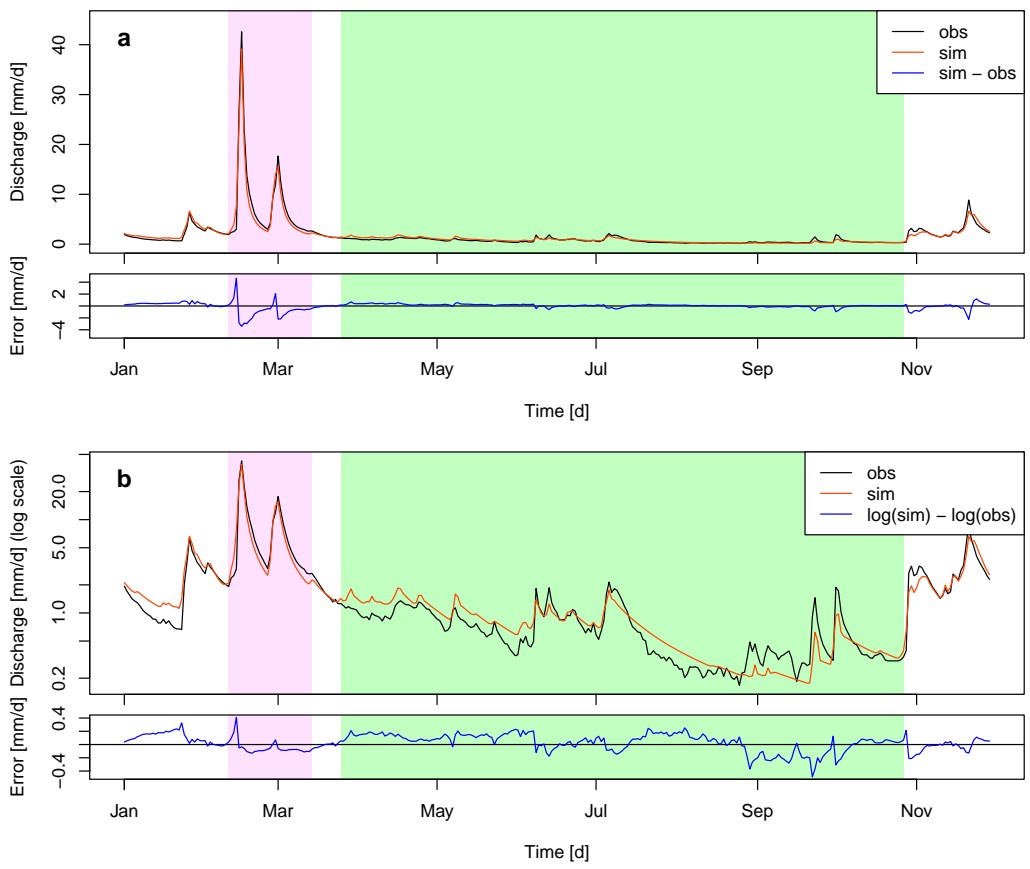

**Figure 1.** Panel a: Observed and simulated streamflow (with the GR4J model calibrated with NSE) time series (left axis) for the Fecht River at Wintzenheim and the related difference (i.e. error, right axis) between 1 January 1990 and 29 November 1990. Panel b: The same observed and simulated streamflow time series plotted with a logarithmic scale and the difference in log-transformed observed and simulated streamflows. In the boxes, a low-flow period is highlighted in green, which shows that the error is minimal with no transformation (panel a) and much higher with a logarithmic transformation (panel b). The opposite is valid for high flows (in pink).

a transformation on flows before optimization". Others only stated that transformations are used "to remove the bias towards
high flows" (Smakhtin et al., 1998), "to fit low flow periods" (Pechlivanidis et al., 2014) or to put "more weight on low flow"
(Garcia et al., 2017).

While Fig. 1 illustrates these assertions to some degree, there is a lack of a general assessment of the impact of transformations on the calculation of criteria over diverse conditions. The objective of this study is to provide new insights to fill this gap in the literature; namely, we aim to perform a systematic evaluation of the impact of 11 streamflow transformations on the
errors made by hydrological models over specific parts of the streamflow ranges. In order to help generalize our results, we
set up a methodology which we applied on a large set of catchments. Here, we will not consider here metrics calculated with

| Reference | $\sqrt{Q}$ | $log(Q)$ | $1/\sqrt{Q}$ | $Q^{-1}$ | Box–Cox | Other power law transformations | Mix |
|---|---|---|---|---|---|---|---|
| Abdulla et al. (1999) | | | | | ✓(10 values between -1 and 1) | | |
| Beck et al. (2016) | | ✓ | | | | | |
| Box and Cox (1964) | | | | | ✓(multiple values between -1 and 1) | | |
| Chapman (1964) | | | ✓ | | | | |
| Chiew et al. (1993) | | | | | | ✓ | |
| Dawdy and Lichty (1968) | | | | | | ✓ | |
| de Vos et al. (2010) | | ✓ | | | | | |
| Ding (1966) | | | ✓ | | | | |
| Duan et al. (2007) | | | | | ✓(basin specific) | | |
| Farmer and Vogel (2016) | | ✓ | | | | | |
| Garcia et al. (2017) | ✓ | ✓ | | ✓ | | | |
| Hogue et al. (2000) | | | | | ✓(0.3) | | |
| Houghton-Carr (1999) | | ✓ | | | | | |
| Huang et al. (2023) | | | | | ✓(multiple values between -1 and 1) | | |
| Ishihara and Takagi (1970) | | | ✓ | | | | |
| Krause et al. (2005) | | ✓ | | | | | |
| Lerat et al. (2020) | | | | | ✓(0.2) | | |
| Nicolle et al. (2014) | | | | | | | ✓ |
| Oudin et al. (2006) | ✓ | ✓ | | | | | |
| Pechlivanidis et al. (2014) | | ✓ | | | | | |
| Pushpalatha et al. (2012) | ✓ | | | ✓ | | | |
| Quesada-Montano et al. (2018) | | ✓ | | | | | |
| Santos et al. (2018) | | ✓ | | | | | |
| Seeger and Weiler (2014) | | ✓ | | | | | |
| Smakhtin et al. (1998) | | ✓ | | | | | |
| Song et al. (2019) | ✓ | ✓ | | | | | |
| van Werkhoven et al. (2008) | | | | | ✓(0.23) | | |
| Vázquez et al. (2008b) | | | | | ✓(0.25) | | |
| Vrugt et al. (2006) | | | | | ✓(0.3) | | |

**Table 1.** Non-exhaustive list of references using streamflow transformations.

specific streamflow selection procedures such as keeping only streamflow values under or over a threshold or the use of relative streamflow.

## 2   Material and method

 ### 2.1   Catchment set and data

We used data from 325 catchments around France (Chauveau et al., 2011) in order to (i) generalize the conclusions drawn from this study (Gupta et al., 2014) and (ii) explore possible links between catchment characteristics and specific behaviours of transformations. These catchments were chosen for the low human impact on the precipitation–streamflow relationship and for the low rate of missing streamflow data ($< 0.5$ %) over the period of interest. Moreover, the catchments are spread throughout France (Figure 2), thus representing a wide variety of meteorological and hydrological conditions.

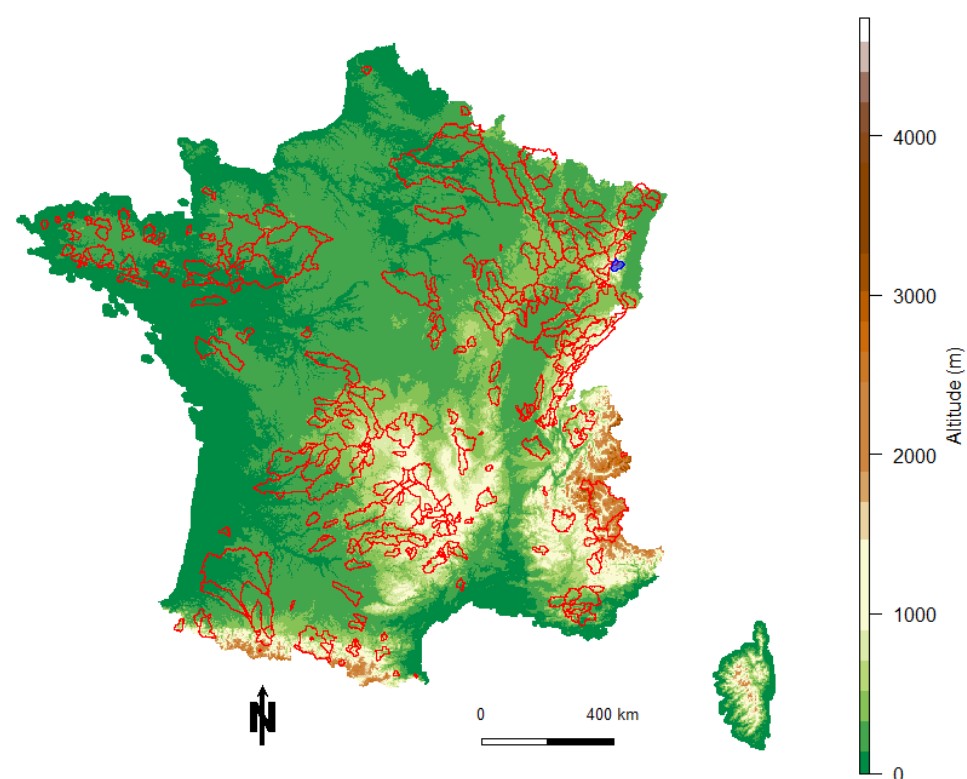

**Figure 2.** Map of France with the location of the 325 catchments used. The Fecht River at Wintzenheim, which is used as an example throughout this paper, is coloured in blue.

Precipitation and temperature data were retrieved from the Météo-France SAFRAN reanalysis (Vidal et al., 2010). Streamflow data were retrieved from the French HydroPortail database (Leleu et al., 2014). Daily meteorological and hydrological data from the period August 1985–July 2005 were used, with August 1985–July 1995 as the calibration period and August 1995–July 2005 as the independent evaluation period. The main characteristics of the 325 catchments are summarized in Table 2, illustrating the large diversity of catchment characteristics encountered, with small to large catchments, various precipitation and temperature conditions, rainfed as well as snowfed catchments, and catchments facing low to high baseflow components. Moreover, the large range of land cover, slope and hydraulic length strengthens the diversity of possible catchment response. This table also shows that the climatic and hydrological conditions are similar between the two periods, with the evaluation period being only slightly warmer and wetter than the calibration period and other indicators showing only slight variations.

| Characteristic | Period | Minimum | Median | Maximum |
|---|---|---|---|---|
| Surface area [km$^2$] | - | 5 | 226 | 13 484 |
| Min. altitude [m a.s.l.] | - | 6.0 | 209.0 | 2 154.0 |
| Median altitude [m a.s.l.] | - | 53.0 | 368.0 | 2 741.0 |
| Max. altitude [m a.s.l.] | - | 93.0 | 784.0 | 3 997.0 |
| Median slope [deg] | - | 1.1 | 7.4 | 35.8 |
| Median hydraulic length [km] | - | 2.1 | 19.0 | 200.7 |
| Artificial land cover [%] | - | 0.0 | 2.1 | 18.2 |
| Agricultural land cover [%] | - | 0.0 | 54.2 | 97.7 |
| Forest land cover [%] | - | 0.0 | 43.5 | 100.0 |
| Mean annual precipitation [mm y$^{-1}$] | Calibration | 651 | 1 009 | 2 204 |
|  | Evaluation | 691 | 1 025 | 2 077 |
| Fraction of solid precipitation [%] | Calibration | 0.3 | 2.5 | 59.1 |
|  | Evaluation | 0.0 | 2.2 | 50.3 |
| Mean air temperature [°C] | Calibration | -1.1 | 10.0 | 13.9 |
|  | Evaluation | -0.9 | 10.3 | 14.2 |
| Mean annual potential evapotranspiration [mm y$^{-1}$] | Calibration | 252 | 661 | 858 |
|  | Evaluation | 267 | 678 | 871 |
| Mean annual runoff [mm y$^{-1}$] | Calibration | 101 | 405 | 2485 |
|  | Evaluation | 123 | 410 | 2250 |
| Baseflow index (BFI) [−] | Calibration | 0.01 | 0.22 | 0.68 |
|  | Evaluation | 0.01 | 0.23 | 0.76 |
| Aridity index [−] | Calibration | 0.03 | 0.33 | 0.74 |
|  | Evaluation | 0.01 | 0.33 | 0.77 |
| Aridity seasonality [−] | Calibration | 0.69 | 1.33 | 1.64 |
|  | Evaluation | 0.62 | 1.36 | 1.72 |
| Centre of mass of annual runoff [doy] | Calibration | 117 | 152 | 248 |
|  | Evaluation | 113 | 145 | 244 |
| Central slope of the flow duration curve [−] | Calibration | 0.39 | 1.05 | 5.05 |
|  | Evaluation | 0.40 | 1.01 | 5.26 |

**Table 2.** Characteristics of the 325 catchments. The minimum, median and maximum columns represent the lowest, $163^{th}$ and highest value over the 325 catchments for every characteristic. The baseflow index values range between 0 and 1, with 1 being the highest value (highest baseflow). The baseflow index was calculated according to Pelletier and Andréassian (2020) with the `baseflow` R package (Pelletier et al., 2021). The aridity index and the seasonality of aridity were calculated according to Knoben et al. (2018), the centre of mass of annual runoff was calculated according to Stewart et al. (2005), and the central slope of the flow duration curve was calculated according to McMillan et al. (2017). Physiographic data were calculated using the SRTM DEM (Farr et al., 2007) and the Corine Land Cover data (Copernicus, 2012). The calibration and evaluation periods are 1985–1995 and 1995–2005, respectively. The maps with sample statistics for these catchment features are included in Appendix C

## 2.2 Hydrological model

The GR4J model is a lumped conceptual daily rainfall–runoff model (Perrin et al., 2003). In this model, the effective precipitation is derived from the reduction in total precipitation by vegetation interception and by evapotranspiration from a soil moisture accounting production store. The effective precipitation is then routed through two unit hydrographs and one routing store. Groundwater exchange can occur from or to neighbouring catchments. A complete description of the model's equations is provided by Perrin et al. (2003).

This model contains four free parameters to calibrate against streamflow observations: the maximum capacity of the production store (X1, in $\mathrm{mm}$), the groundwater potential exchange (X2, in $\mathrm{mm\,d^{-1}}$), the 1-day-ahead routing store capacity (X3, in $\mathrm{mm}$) and the time characteristics of the unit hydrographs (X4, in $\mathrm{d}$).

For the catchments with a proportion of solid precipitation (considered here as precipitation occurring with negative air temperatures) greater than 10 % of the total precipitation, a snow model, CemaNeige, was used. This model is based on a degree–day approach and comprises two parameters to calibrate: the melt rate coefficient ($K_f$, in $\mathrm{mm\,^{\circ}C^{-1}\,d^{-1}}$) and a parameter regulating the energy of the snowpack ($c_T$, dimensionless). In order to consider intra-catchment variability, CemaNeige was applied to five elevation bands of equal area, which makes it possible to account for temperature and precipitation gradients (see Valéry et al., 2014, for more details).

In this work, we also use the GR6J model (Pushpalatha et al., 2011) to assess the transferability of the conclusions drawn. GR6J adds two parameters to GR4J: X5 [-], which enables an inversion of the direction of the groundwater exchange throughout the year, and X6 [mm], which is the maximum capacity of an additional exponential store, whose purpose is to improve low-flow simulations.

All the calculations are made with the airGR R package (Coron et al., 2017, 2022). The built-in optimization algorithm, an initial parameter grid screening followed by a steepest gradient approach, is chosen due to its known satisfactory performance with the GR models (Perrin et al., 2003; Mathevet, 2005; Coron et al., 2017). All optimization criteria and streamflow transformations used in this work are embedded in airGR.

## 2.3 Objective functions

In order to assess the impact of transformations, the hydrological models are calibrated with several objective functions over the 1985–1995 period. However, in order to estimate how transformations impact the simulated time series, the 1995–2005 independent evaluation period is also used. In both cases, a 1-year spin-up period preceding the aforementioned periods is used.

Three objective functions are chosen for their wide use in calibrating hydrological models: the well-known Nash–Sutcliffe efficiency (NSE; see Nash and Sutcliffe, 1970), the Kling–Gupta efficiency (KGE; see Gupta et al., 2009) and the modified Kling–Gupta efficiency (KGE'; see Kling et al., 2012). The NSE concentrates most of the analyses of this work and the KGE

and KGE' objective functions are used to assess the generality of the results. These three criteria are detailed in Equations 1, 2 and 3.

$$E_{NSE} = 1 - \frac{\sum_{t=1}^{N}(Q_t^s - Q_t^o)^2}{\sum_{t=1}^{N}(\overline{Q^o} - Q_t^o)^2} \tag{1}$$

$$E_{KGE} = 1 - \sqrt{(r-1)^2 + (\frac{\overline{Q^s}}{\overline{Q^o}} - 1)^2 + (\frac{s_d(Q^s)}{s_d(Q^o)} - 1)^2} \tag{2}$$

$$E_{KGE'} = 1 - \sqrt{(r-1)^2 + (\frac{\overline{Q^s}}{\overline{Q^o}} - 1)^2 + (\frac{C_V(Q^s)}{C_V(Q^o)} - 1)^2} \tag{3}$$

where $N$ is the total number of days of the test period, $Q_t^s$ and $Q_t^o$ are the simulated and observed streamflows, respectively, at time step t, $\overline{Q^o}$ (resp. $\overline{Q^s}$) is the average observed (resp. simulated) streamflow over the period, $r$ is the correlation coefficient, $s_d$ is the standard deviation and $C_V$ is the coefficient of variation.

## 2.4 Streamflow transformations

The hydrological models are calibrated by applying different transformations to streamflow values in the calculation of the objective functions. Nine to 11 transformations are used (Table 3), as well as three objective functions. In addition to the transformations mentioned in the Introduction, four additional transformations are used. The squared ($Q^2$) transformation is applied, as this can be used for focusing on floods (Tan et al., 2005), and its inverse ($Q^{-2}$) is applied, as it focuses on low flows. Furthermore, two composite criteria, $\frac{f(Q)+f(Q^{-1})}{2}$ and $\frac{f(Q)+f(log(Q))}{2}$ (with $f$ standing for NSE, KGE or KGE'; Nicolle et al., 2014), are added since they can be used as a compromise between criteria focusing on ranges of streamflows that are too specific. The two transformations containing the use of a logarithm are not applied to KGE and KGE', as they cause numerical instabilities and unit-dependence, as shown by Santos et al. (2018). Regarding the Box–Cox transformation, equation 10 by Santos et al. (2018) is used to avoid the same issues as for the logarithmic transformation with a $\lambda$ value equal to 0.25, as suggested by Vázquez et al. (2008a).

## 2.5 Evaluation methodology

In order to evaluate the impact of the transformations on model calibration, we use a common analysis framework that aims at analysing the behaviour of transformations at every simulation time step. The general methodology, which is applied for each catchment and for each objective function, is detailed here and in Fig. 3 (for illustrative purposes for only two transformations):

1. the hydrological model is calibrated against observed streamflows for a catchment and with a given objective function, successively with different transformations (see Fig. 3a for two transformations only),

**Table 3.** The 11 transformations used in this study and the criteria they are applied to. The abbreviations provided here are used in the figures and text. The $\lambda$ value for the Box–Cox transformation is 0.25.

| Transformation | Abbreviation | NSE | KGE | KGE' |
|---|---|---|---|---|
| $Q^2$ | 2 | ✓ | ✓ | ✓ |
| - | 1 | ✓ | ✓ | ✓ |
| $\sqrt{Q}$ | 0.5 | ✓ | ✓ | ✓ |
| $Q^{0.2}$ | 0.2 | ✓ | ✓ | ✓ |
| Box–Cox | $boxcox$ | ✓ | ✓ | ✓ |
| $\frac{f(Q)+f(log(Q))}{2}$ | $QlogQ$ | ✓ | | |
| $\frac{f(Q)+f(Q^{-1})}{2}$ | $QinvQ$ | ✓ | ✓ | ✓ |
| $log(Q)$ | $log$ | ✓ | | |
| $1/\sqrt{Q}$ | $-0.5$ | ✓ | ✓ | ✓ |
| $Q^{-1}$ | $-1$ | ✓ | ✓ | ✓ |
| $Q^{-2}$ | $-2$ | ✓ | ✓ | ✓ |

2. for each time step, the absolute error $|Q_t^s - Q_t^o|$ is calculated for the simulations obtained with the nine (or 11) transformations (see Fig. 3b for two transformations only),

3. these daily absolute errors are ranked from the smallest to the largest among the nine (for KGE or KGE') or 11 (for NSE) simulations (see Fig. 3c for two transformations only),

4. the time series of daily ranks are sorted according to the sorted observed streamflow time series (see Fig. 3d for two transformations only),

5. the sorted ranks are aggregated over 200 sequential intervals of an equal number of time steps to smooth the results and facilitate the visual analysis. Two aggregations were made:

   – extraction of the transformation with the most 'number-1' ranks (see Fig. 3e for two transformations only),

   – calculation of the average rank for each class (see Fig. 3f for two transformations only).

The use of ranks to classify the proximity between model simulations and streamflow observations could be criticized, since it gives the same importance to large and small errors. This option was preferred to the use of direct (normalized) errors. Indeed, ranks make it possible to consider together various flow ranges, where the magnitude of errors can be very different. The impact of this methodological choice will be discussed in section 3.4.

The methodology above is applied catchment by catchment. Then, to aggregate results over the 325 catchments, we either identify the transformation with the most number-1 ranks or average the ranks over the 325 catchments.

When modellers choose an objective function (or, if relevant, a transformation), the main objective is to have a model fit for purpose, e.g. to be the best for low flows if the target is low flows. In the following, we evaluate the link between the

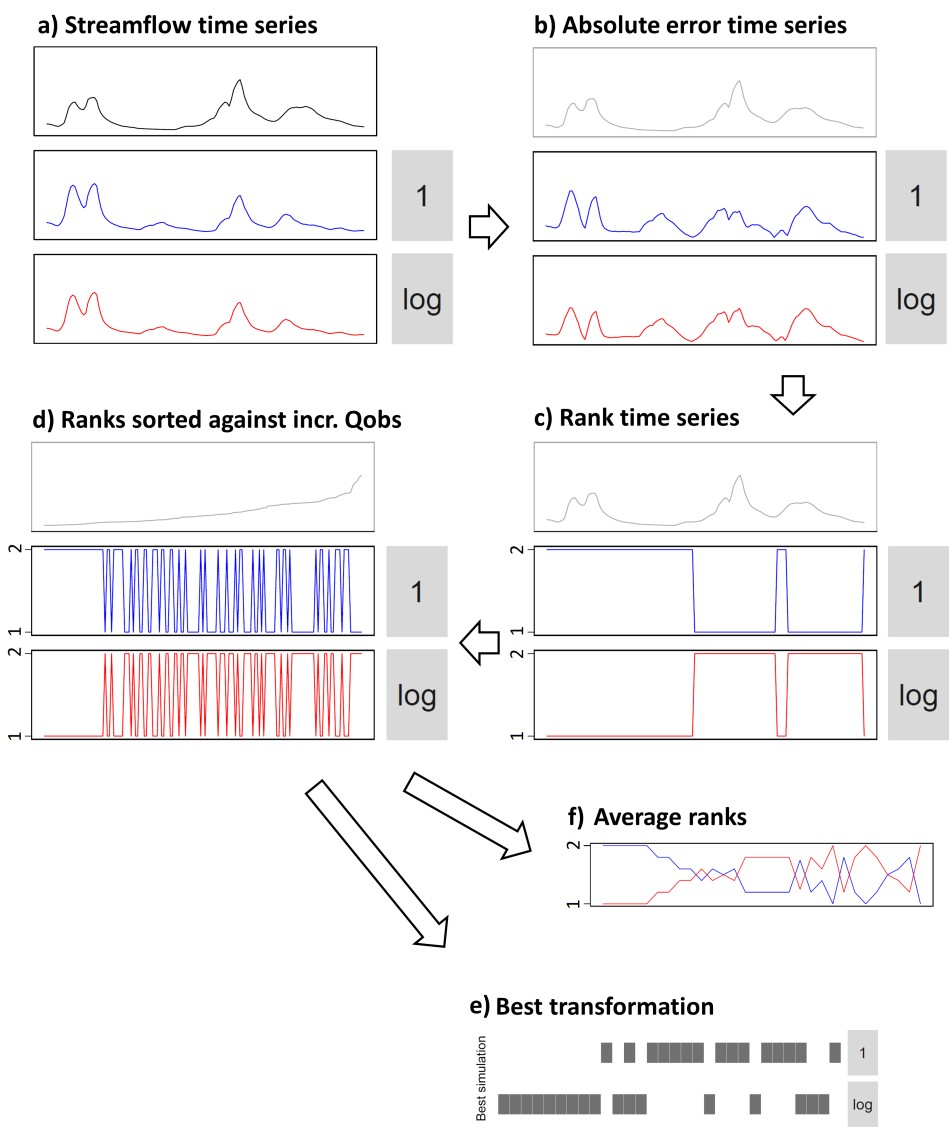

**Figure 3.** General methodology applied to assess the impact of transformations on the diverse ranges of streamflows. Here the example is shown over a short period and for only two transformations and a single catchment. In the study, the methodology is applied for nine to 11 transformations, 10-year periods and 200 intervals. a) Observed and simulated streamflow time series; b) absolute values of errors in transformed streamflow time series; c) ranking of error time series; d) sorting of ranked time series according to increasing observed streamflows. For the next subplots, results are aggregated over intervals: e) identification of the transformation with most number-1 ranks for each interval; f) calculation of average rank for each transformation and each interval. See the Methods section for more details.

objective function and transformation selected and the accuracy of the model using the 200 flow intervals described above. We successively performed this analysis on

- the calibration period over a single catchment with GR4J calibrated on NSE,

- the calibration and evaluation periods over the 325 catchments with GR4J calibrated on NSE,

- the evaluation period over the 325 catchments with GR6J calibrated on NSE and GR4J calibrated on KGE.

We also tried to link the results to the characteristics of the 325 catchments over the evaluation period with GR4J calibrated on NSE, using the Spearman correlation. Last, we questioned the methodology used to compare transformations for the calibration period over the 325 catchments with GR4J calibrated on NSE.

## 3 Results and discussion

### 3.1 Analysis of the impact of transformations for a specific catchment

Figure 4 illustrates an example application of the methodology for a single catchment, the Fecht at Wintzenheim, for the GR4J model calibrated with the NSE objective function and for 11 different transformations. Here we show which transformation leads to the most number-1 ranks for each of the 200 intervals, for low flows (on the left) to high flows (on the right). It appears that some transformations are often ranked first (such as the $-2$, $-1$, 1 and 2 transformations). Conversely, some transformations are rarely or never ranked first (such as $QlogQ$ or 0.2). In this figure, the transformations are represented in an order from a presupposed good representation of high flows (top row) to a presupposed representation of low flows (bottom row), because except for composite transformations, the aforementioned transformations are presented in an order of decreasing power. We can see that the logic is respected quite well, with transformations 2 and 1 being very well represented regarding intervals corresponding to high flows, and transformations $-2$ and $-1$ being very well represented regarding intervals corresponding to low flows. This does not preclude some transformations from being identified as the best one (or equally the best one, as ties are represented in Fig. 4) for unexpected intervals, such as transformation 2 that shows good results for some low-flow categories.

A second representation is given in Fig. 5 with the average rank of transformations (i.e. the average of the ranks of transformations over all the time steps of the interval concerned) for each of the 200 intervals. In this figure, we see that the transformations remain rather close together for low flows, with an average rank between 5 and 7. By contrast, the spread is larger for high streamflows with average ranks between 4 and 9. Specifically, several transformations share the best average rank values for low flows, such as the $-1$, $log$ and $-0.5$ transformations. Interestingly, the $-2$ transformation, which is supposedly the transformation giving the highest weight to low flows and was identified as the transformation with the most number-1 ranks for a high number of intervals in Fig. 4, only shows the best average rank for the very first interval, and then quickly shows a much worse average rank. This might indicate that the $-2$ transformation gives a high weight to errors over a limited number of time steps with the lowest streamflows (see Figure B1 in the Appendix for further analyses).

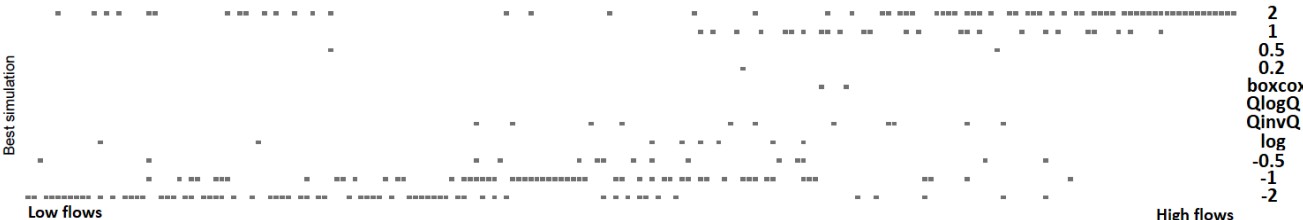

**Figure 4.** Identification of the simulation with the most number-1 ranks for 200 intervals ordered by increasing observed streamflows. Example for the Fecht River at Wintzenheim, over the calibration period (1985–1995), for the GR4J model calibrated with NSE. The CemaNeige model was not used. Each rectangle identifies for one interval which transformation(s) provides the most number-1 ranks, including ties. The $\lambda$ value for the Box–Cox transformation is 0.25.

Regarding the middle range of streamflows, a couple of transformations show the best average rank, such as the $-1$, $log$ and $-0.5$ transformations, but also progressively, as streamflows get higher, the $0.2$, $QinvQ$ and $boxcox$ transformations. Interestingly, this indicates that while being quite average most of the time (analysis not shown here), these transformations

still have better average ranks than transformations with more occurrences of rank 1. Finally, regarding high flows, the 2, 1, $QlogQ$ and $0.5$ transformations take the lead.

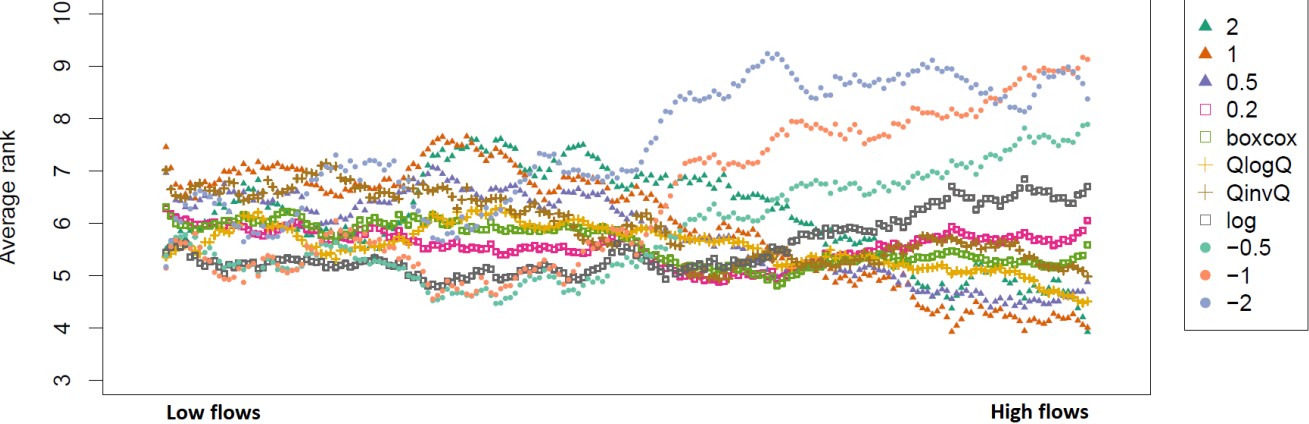

**Figure 5.** Average rank for each transformation. Example for the Fecht River at Wintzenheim, over the calibration period (1985–1995), for the GR4J model calibrated with NSE. A smoothing window (10-value moving average) is applied to improve legibility. The $\lambda$ value for the Box–Cox transformation is 0.25.

### 3.2 Analysis of the impact of transformations for the 325 catchments

#### 3.2.1 Analysis of the calibration period

While some trends could be identified in the analysis of a single catchment in the previous section, the results are impacted by a rather high level of noise for successive intervals. To circumvent this issue, and to generalize the results, we perform a similar analysis over the 325-catchment set presented in section 2. This analysis is shown in Fig. 6 with the GR4J model calibrated with the NSE objective function. Results are presented for the calibration period. In this figure, the best simulation is identified for each catchment and for each interval according to the methodology presented in Fig. 3f. Then, for each interval, the simulation with the most number-1 ranks among the 325 catchments is labelled as the best. A clear pattern appears: The 2 transformation is the best for high flows, and the 1 transformation is the best for slightly lower flows. Regarding low flows, the best transformation for the most extreme flows is the $-2$ transformation, followed by the $-1$ and $-0.5$ transformations. This result confirms that the goal of transformations, which is to distort the streamflow time series, is easily reached when used for calibration. The only surprise is that only 5 out of the 11 transformations are identified as the best for at least one interval. However, the present analysis is binary and could result in a more precise diagnosis being missed.

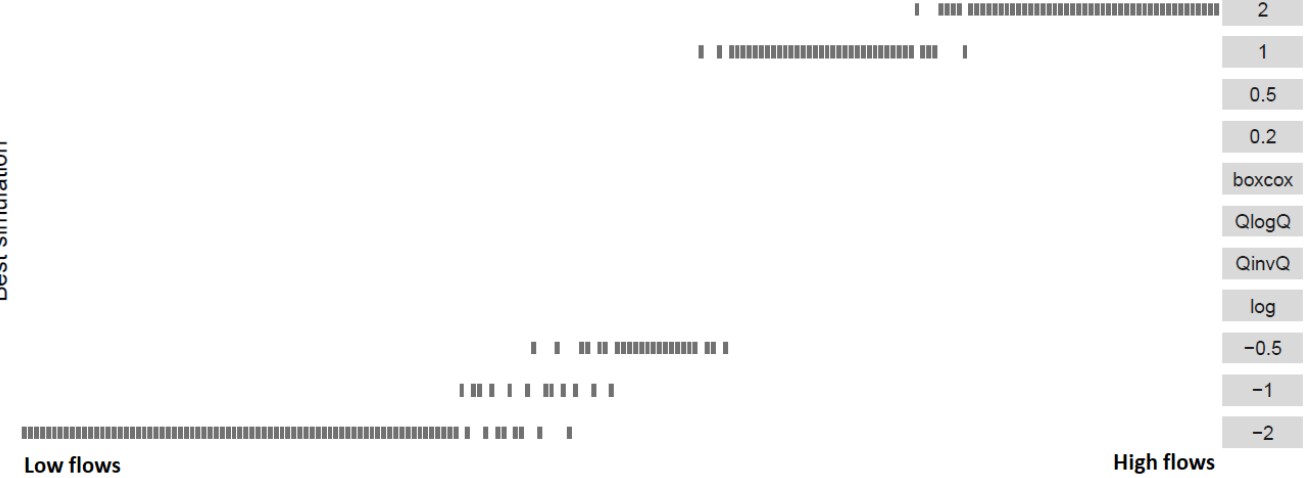

**Figure 6.** Identification of the simulation with the most number-1 ranks for the 325 catchments and for 200 intervals ordered by increasing observed streamflows. To provide this analysis, the output of Fig. 3f is used, and for each interval, the total of number-1 ranks is cumulated over the 325 catchments to identify the simulation with the most number-1 ranks. The GR4J model is used and is calibrated with NSE. Results are shown for the calibration period. The $\lambda$ value for the Box–Cox transformation is 0.25.

In order to better understand the behaviours of the different transformations, we show in Fig. 7 the interval-averaged rank of all transformations. The best average rank most of the time is between 4 and 5, except for high flows where it can reach 3.5 (the best being 1). We see that no transformation is always the best, even though some show a rather high interval-averaged rank throughout most of the intervals. Regarding the worst transformations, they show an interval-averaged rank around 8 to 9

out of 11; however, transformation 2 is clearly the worst transformation for low flows and $-2$ is the worst one for high flows. Some general features can be observed. First, several transformations take the lead for low flows: the $-2$ transformation shows the best average rank for the very first intervals but quickly shows a worse average rank. The other leading transformations (i.e. with best average rank), when going from the lowest flows to increasing flows, are successively transformations $-1$, $-0.5$, $log$, $boxcox$, $0.5$ and finally $1$. It is noteworthy that despite being identified as an excellent transformation for high flows in previous figures, transformation 2 is never the best transformation on average. This stems from the fact that even though it is the best high-flow transformation for many catchments, when it is not, its rank is rather low; we could qualify this transformation as an all-or-nothing transformation. Regarding the shape of the curves, we can distinguish four groups. First, transformations 2, 1 and 0.5 show a decreasing curve, i.e. they have a best rank for high flows than for low flows. Conversely, transformations $-2$, $-1$ and $-0.5$ show an increasing curve, i.e. they have a better rank for low flows than for high flows. Transformations $QlogQ$ and $QinvQ$ show the best ranks both for high and low flows, with the worst ranks for the medium range of streamflows (arch-shaped curve). Transformations 0.2 and $log$ show the best ranks for the medium range of streamflows, and the lowest ranks for high and low flows (U-shaped curves). Finally, transformation $boxcox$ shows the best rank for medium to high flows, but not for the highest flows.

Averaging the interval-averaged ranks over the 200 intervals provides an overview of the general ranking of the transformations (Table 4). This analysis leads to the following ranking: transformations 0.2, $log$ and $boxcox$ have the lowest (i.e. best) average rank, followed by 0.5, $QlogQ$, $-0.5$, $QinvQ$, 1, $-1$, $-2$ and 2. Typically, only one transformation is used by modellers for their application. If their application is very specific, this might make sense, as it is possible to identify transformations that outperform others. However, when their application is multi-purpose, it is very likely that the transformation that is chosen only fits a limited range of streamflows. This is even more striking when we observe that the commonly used 1 transformation is very often applied for calibration despite being the best transformation for only a very limited portion of streamflow range, i.e. high flows. In addition, the transformations that show the best average rank are not the most widely used in the literature (0.2, $log$ and $boxcox$).

The impact of using 200 instead of 100 or 500 intervals is illustrated in Appendix A, which shows that applying 200 intervals is a good compromise between too-coarse information and too-noisy information. It also shows that the results are only marginally impacted. We will therefore keep this number of intervals in the following. In addition, we will now only use figures similar to Fig. 7 (i.e. the evolution of average ranks through the range of streamflows), as it constitutes an efficient way of visualizing results and provides enough information to understand the behaviours of the transformations.

### 3.2.2 Are conclusions transferable to an independent period?

In the previous section, the results were presented for the calibration period, i.e. in optimal conditions to understand how transformations impact model simulations when used for model calibration. However, the purpose of using models is to apply them on periods different from those used for calibration. Here we show the average range for the GR4J model calibrated with NSE over the evaluation period. The objective is to discuss whether the conclusions drawn for the calibration period are transferable to this independent period. The results are shown in Fig. 8.

**Table 4.** Interval-averaged ranks for a calibration of the GR4J model performed with NSE. Results are shown for the calibration period. The $\lambda$ value for the Box–Cox transformation is 0.25.

| Transformation | Average rank |
|:---:|:---:|
| 2 | 7.49 |
| 1 | 6.24 |
| 0.5 | 5.50 |
| 0.2 | 5.23 |
| *boxcox* | 5.25 |
| *QlogQ* | 5.59 |
| *QinvQ* | 6.12 |
| *log* | 5.30 |
| $-0.5$ | 5.80 |
| $-1$ | 6.39 |
| $-2$ | 7.08 |

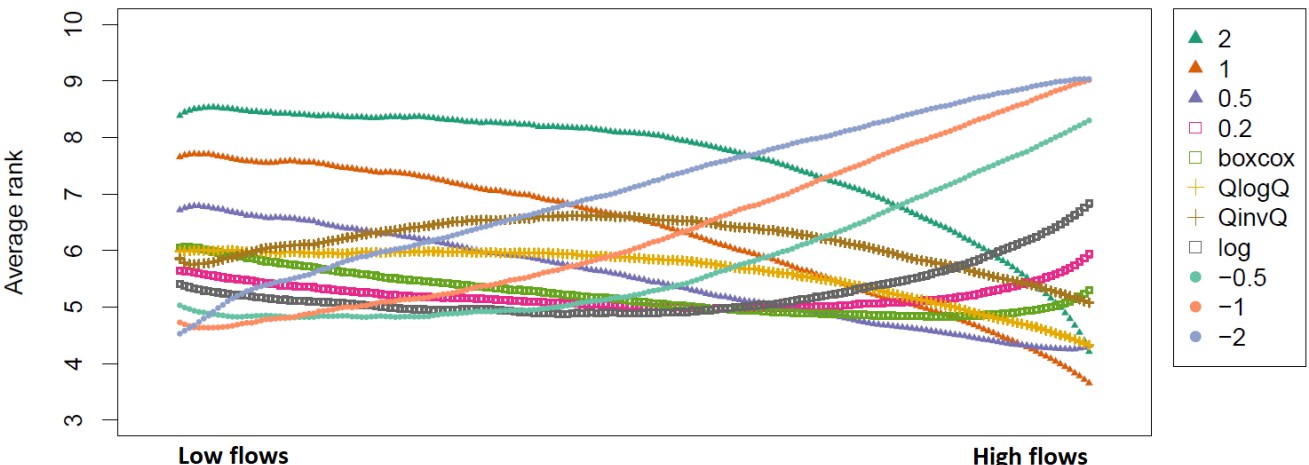

**Figure 7.** Interval-averaged rank over the 325 catchments and for 200 intervals ordered by increasing observed streamflows. To provide this analysis, the output of Fig. 3f is used, and for each interval, the mean rank is averaged over the 325 catchments. The GR4J model is used and is calibrated with NSE. Results are shown for the calibration period. A smoothing window (10-value moving average) is applied to improve legibility. The $\lambda$ value for the Box–Cox transformation is 0.25.

Interestingly, the results are very similar to those of Fig. 7. The main discrepancy is that the average rank of the best transformation shows a higher value for the evaluation period than for the calibration period, and correspondingly the average rank of the worst transformation shows a lower value for the evaluation period. In other words, over the evaluation period, the transformations lead to simulations that are less specific, i.e. closer to each other. For the lowest flows, the transformations *log*

and $0.2$ show the best average ranks, while for higher flows, transformations $boxcox$, $0.5$ and $1$ successively lead the pack. Transformations $2$ and $-2$ are never at the top of the average ranks, indicating that in addition to being an all-or-nothing option for calibration, they are also poorly transferable to an independent period, even for their respective range of expertise. The averaging of the average ranks over the 200 intervals is shown in Table 5. This leads to the following ranking: transformations $0.2$, $boxcox$ and $log$ have the lowest average rank, followed by $0.5$, $QlogQ$, $-0.5$, $QinvQ$, $-1$, $1$, $-2$ and $2$. Compared to the calibration period, this ordering is only marginally modified and the average ranks over the 200 intervals are only slightly different. This indicates that, while not significantly modifying the conclusions of the previous analysis, over an independent period, the transformations lose specificity for their presupposed range of expertise so as to gain performance for the rest of the streamflow range.

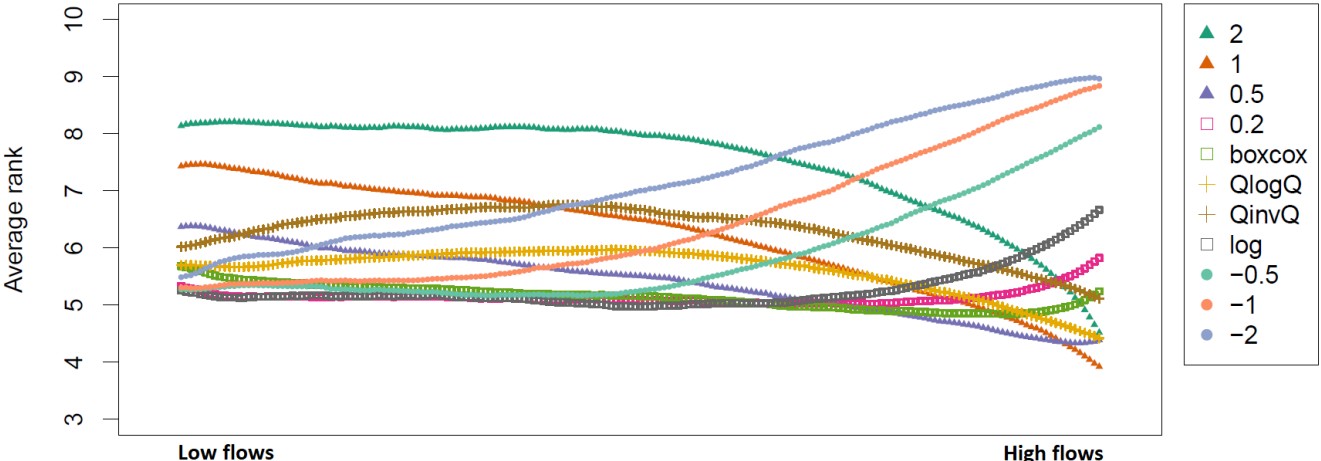

**Figure 8.** Average rank over the 325 catchments and for 200 intervals ordered by increasing observed streamflows. To provide this analysis, the output of Fig. 3f is used, and for each interval, the mean rank is averaged over the 325 catchments. The GR4J model is used and is calibrated with NSE. Results are shown for the independent evaluation period. A smoothing effect (10-value moving average) is applied to improve legibility. The $\lambda$ value for the Box–Cox transformation is 0.25.

### 3.2.3   What is the impact of the choice of objective functions and the hydrological models?

All previous analyses were led with the GR4J model calibrated with NSE. In the following, we assess the impact of using another hydrological model, GR6J, as well as two additional objective functions, KGE and KGE'. Although this model and these objective functions can be considered to be not drastically different from GR4J and NSE, we believe that they provide useful transferability information for the results. Indeed, these two new objective functions are increasingly reported in the literature, which justifies their use. The following analyses are made for the independent evaluation period only.

Figure 9 provides the interval-averaged ranks of the 11 transformations for the GR6J model calibrated with NSE, assessed over the independent evaluation period. The general shape of this plot is rather similar to that of Fig. 8. The main differences

**Table 5.** Interval-averaged ranks for a calibration of the GR4J model performed with NSE. Results are shown for the independent evaluation period. The $\lambda$ value for the Box–Cox transformation is 0.25.

| Transformation | Average rank |
|:---:|:---:|
| 2 | 7.44 |
| 1 | 6.19 |
| 0.5 | 5.40 |
| 0.2 | 5.16 |
| *boxcox* | 5.16 |
| *QlogQ* | 5.59 |
| *QinvQ* | 6.26 |
| *log* | 5.29 |
| −0.5 | 5.88 |
| −1 | 6.45 |
| −2 | 7.16 |

concern transformation 2, which appears to show much worse ranks than the other transformations for GR6J than for GR4J, especially for low flows. This might stem from the fact that compared to GR4J, GR6J was mainly developed for improving low flows and therefore contains two more parameters to optimize, which focus on low-flow-generating processes and consequently could be less identifiable with transformation 2. We can also see that for most of the intervals (i.e. for the main part of the streamflow range), the best transformation is better identified for GR6J than for GR4J, as the latter shows very close curves most of the time. Table 6 presents the values for the interval-averaged ranks. This table shows that the interval-averaged ranks are very similar for the two models except for transformation 2. We can therefore conclude that for the models used here, the relative performance of transformations is very similar across the streamflow range

Figure 10 shows how the interval-averaged ranks of transformations evolve when we use different objective functions. Interestingly, these two panels are similar to each other and to Fig. 8 (although the reader should note the absence of the two log-dependent transformations for KGE and KGE'). This means that the use of transformations seems to lead to similar ranges of streamflows that are targeted by calibration whatever the objective function used, in particular for the very common NSE, KGE and KGE'. Here we do not show the interval-averaged ranks over the 200 intervals, as the number of transformations differs between NSE on the one hand and KGE and KGE' on the other hand.

### 3.3 Links between catchment characteristics and transformations

We tried to identify links between catchment characteristics and the performance of transformations to better understand the behaviour of the different transformations but also to potentially help prescribe transformations knowing the catchment characteristics. To do so, we used the Spearman correlation to analyse how the interval-averaged ranks of transformations for each catchment could relate to the catchment characteristics listed in Table 2. The GR4J model calibrated with NSE was used

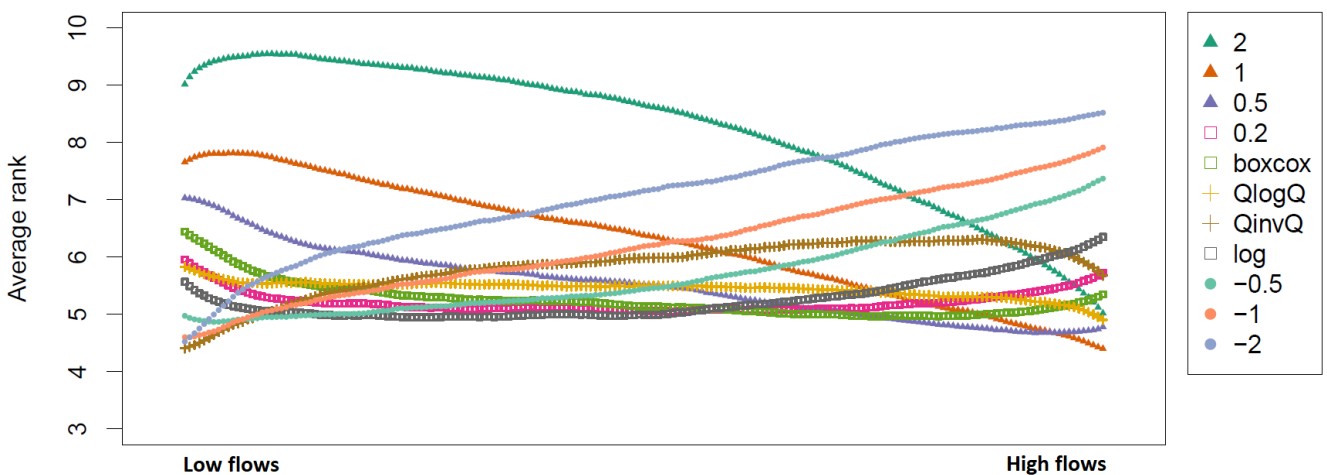

**Figure 9.** Same as Fig. 8 for the GR6J model calibrated with NSE. The $\lambda$ value for the Box–Cox transformation is 0.25.

**Table 6.** Interval-averaged ranks for a calibration of the two hydrological models performed with NSE. Results are shown for the independent evaluation period. The $\lambda$ value for the Box–Cox transformation is 0.25.

| Transformation | GR4J | GR6J |
|:---:|:---:|:---:|
| 2 | 7.44 | 8.15 |
| 1 | 6.19 | 6.28 |
| 0.5 | 5.40 | 5.54 |
| 0.2 | 5.16 | 5.25 |
| *boxcox* | 5.16 | 5.29 |
| $QlogQ$ | 5.59 | 5.44 |
| $QinvQ$ | 6.26 | 5.79 |
| $log$ | 5.29 | 5.29 |
| $-0.5$ | 5.88 | 5.71 |
| $-1$ | 6.45 | 6.23 |
| $-2$ | 7.16 | 7.04 |

here. In order to maximize the possibility of identifying strong links, only the catchment characteristics and interval-averaged ranks over the calibration period were used.

The most important correlations were found between the BFI values and the $QlogQ$ (correlation equal to 0.56), *boxcox* (0.51), 0.2 (0.50) and $log$ (0.38) transformations. Negative correlations were found between BFI and the $-1$ ($-0.32$) and $-0.5$ ($-0.30$) transformations. It means that the lower the BFI, the better the use of transformations giving intermediate weight between high and low flows. Conversely, the higher the BFI, the better the use of transformations that give a large weight to low flows. We observed that the central slope of the flow duration curve shows correlations of the same order of magnitude, but

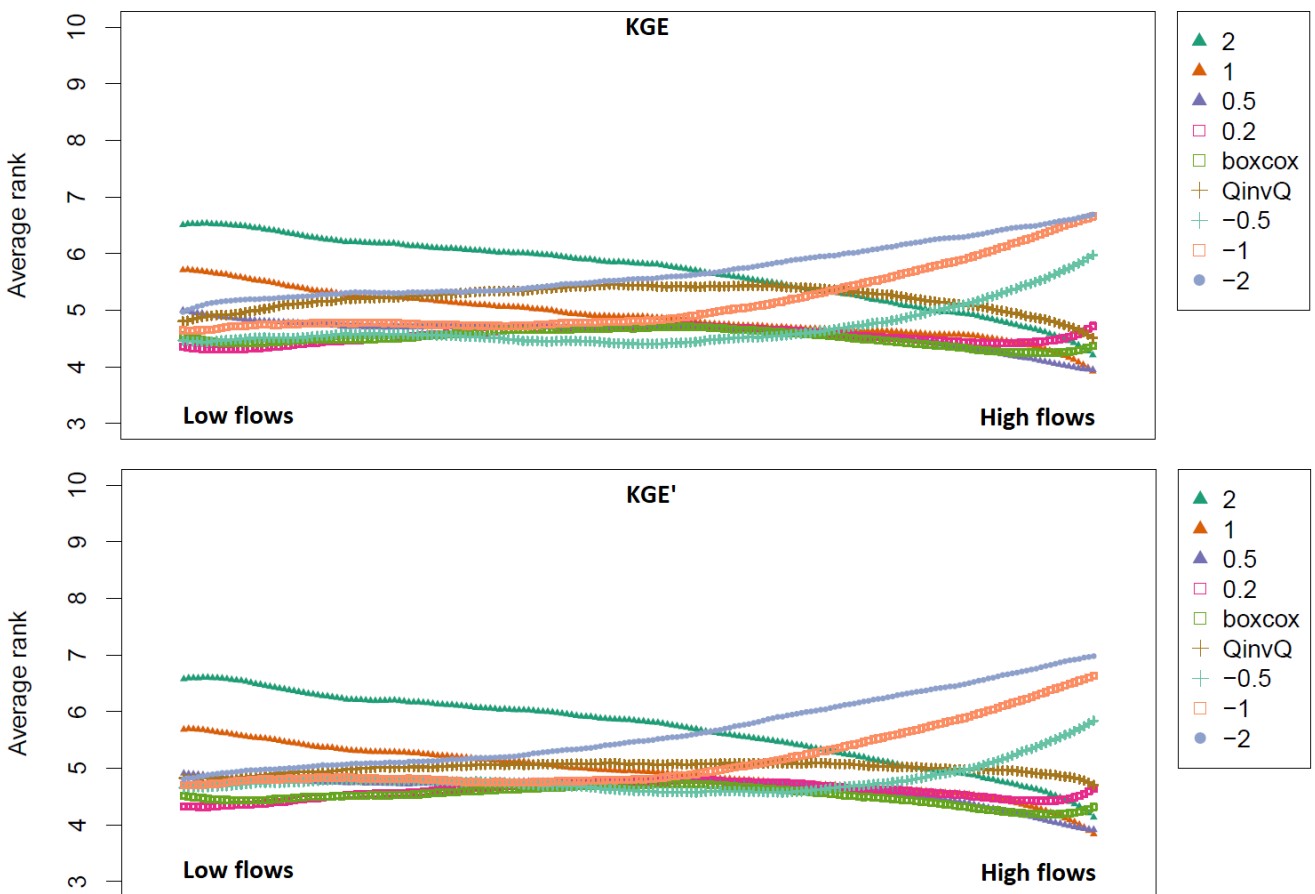

**Figure 10.** Same as Fig. 8 for the GR4J model and the independent evaluation period calibrated with KGE and KGE'. Note that the two transformations using a $log$ transformation were not used with the KGE and KGE' objective functions. The $\lambda$ value for the Box–Cox transformation is 0.25.

opposite to the one shown by the BFI, with exactly the same transformations. This suggests a strong correlation between these indicators. All the other catchment characteristics show no correlation over 0.30 or under $-0.30$, signifying a weak explanation of the performance of transformations with these characteristics. Using the stats::cor.test() function in R, we found that the correlations lower than $-0.11$ and higher than 0.11 were all significant (i.e. p-values $< 0.05$), whereas none of the others were significant. Regarding BFI and the central slope of the flow duration curve, although the above-mentioned correlation values are interesting, they are not sufficient to permit any prescription to be made regarding the choice of transformations for a specific kind of catchment. To deepen this analysis, a co-inertia analysis (Dolédec and Chessel, 1994; Dray et al., 2003) was undertaken on two principal component analyses made, respectively, for the table of catchment characteristics and the table

of transformations; however it could not show any further informative interdependence between catchment characteristics and transformations.

## 3.4 Impact of the method used to compare transformations

In this work, the transformations have been compared using ranks. The idea behind choosing to work with ranks, instead of direct (normalized) errors was to (i) be less impacted by different orders of errors magnitudes between catchments or ranges of streamflows, and to (ii) answer the question of which transformations are the best ones, rather than how good the transformations are. Assigning ranks can, however, have the effect that a rank difference of 1 can signify a small error or a larger one. Nevertheless, ranks are accounted for time step by time step, meaning that if differences between two simulations are very low, there can be changes in the order quite easily, which then results in similar average ranks over the intervals.

The impact of using absolute differences rather than ranks to compare transformations has been assessed in Thirel et al. (2023). The general shapes of curves obtained with this alternative method were found to be similar, although some discrepancies were observed. The large errors of some outlier transformations made it difficult to identify differences between the other transformations. While this could lead to the conclusion that these transformations can be used interchangeably because they seem to lead to very similar errors, it is important to note that the very high normalized MSE of one or two transformations leads to smaller differences for the other transformations. In other words, the results using MSE are impacted by the large error of some transformations and lead to less informative results. We therefore believe that the methodology based on ranks used in this study provides more readable results.

## 3.5 Discussion

The results of this study confirmed some common hypotheses in the literature about efficiency criteria but also provided specific insights that should be helpful for modellers; namely, we confirmed that, as mentioned by diverse authors, transformations can be used "to remove the bias towards high flows" (Smakhtin et al., 1998), "to fit low flow periods" (Pechlivanidis et al., 2014) or to put "more weight on low flow" (Garcia et al., 2017). In addition, we showed that the $log$ or $0.2$ transformations could lead to a better representation of low flows, as stated by Krause et al. (2005) or Chiew et al. (1993).

However, we also added some knowledge to the existing literature. First, we showed that using no transformation of streamflows leads to a performance among the worst for a large range of streamflows (from low to medium range). As using no transformation is still the most widespread way of calculating criteria, this could impact how modellers calibrate models or evaluate them. In addition, we showed that using the most extreme transformations for calibration, both for high (2) and low $(-2, -1)$ flows, leads to a narrow range of streamflows for which the simulations seem satisfactory. This reinforces the idea of trying to define well the purpose of the modelling chains developed and choose the adapted transformation for model calibration. In addition, the fact that it is difficult to identify a transformation leading to the best simulations overall, or simply to sufficiently high performance for the whole range of streamflows, indicates that developing generic models fitting all purposes is still a challenging task for modellers.

## 4 Conclusions

This study explored the impact of mathematical transformations applied to streamflow time series prior to the computation of objective functions for the calibration of hydrological models. Such transformations are often used to focus on different ranges of streamflows, but their actual impact has rarely been assessed. Using the GR4J rainfall–runoff model and 11 transformations for the Nash–Sutcliffe efficiency criterion, we analysed the impact of transformations on streamflow simulations in terms of the difference from observations. We ranked the 11 transformations for each time step and then aggregated the results at different scales. This first analysis on the Fecht River at Wintzenheim showed that, in general, the transformations indeed have the best ranks for the range of streamflows they are presupposed to focus on (e.g. the squared transformation focuses on high flows and shows good ranks for high flows). However, it was shown that some extreme transformations (squared and its inverse) were rather binary, i.e. were either very highly ranked or very poorly ranked, resulting in average ranks not being among the best for most of the streamflow range. In addition, the results also showed that some transformations can have a satisfactory performance for a range of streamflows they are not aimed at and with no clear reason, justifying further analyses with a larger set of 325 catchments.

This larger set of catchments allowed us to generalize the results and to smooth the transformation-ranking relationship. The analysis showed that only a few transformations were identified as being most frequently the best over the 325 catchments, with transformation $-2$ being the best for low flows, followed by transformations $-1$ and $-0.5$, while transformations $1$ and $0.5$ were most often the best ones for high flows. Complementary to this binary analysis, the calculation of the averaged ranks over the 325 catchments showed that the $-2$ transformation is only the best for very low streamflows, meaning that for many catchments, it is often a poorly performing transformation even for rather low flows. Correspondingly, the $2$ transformation was found to be only efficient for very high flows. Some more intermediate transformations, such as the $0.2$, $log$ and $boxcox$ transformations, seem to be less specific but well-performing transformations, quite often being among the best transformations for high, medium and low flows.

Although first tested for the Nash–Sutcliffe criterion objective function, with the GR4J model and over the calibration period, this analysis was performed for two additional objective functions, for one additional hydrological model and for the independent evaluation period. The results were only slightly modified, strengthening the analysis; however, models with different process complexity or other objective functions should be investigated.

The results of this study may have important implications for hydrological modellers. They show that, although some common beliefs about the impact of transformations are confirmed by this study, no a priori assumption on streamflow transformations can be taken as warranted. In fact, some transformations that are focused on extreme ranges of streamflows are shown to lead to calibrated models that are indeed better for these ranges over the calibration period, but that are poorly robust, i.e. that no longer necessarily perform well for this range of streamflows for an independent evaluation period. This might stem from the fact that these transformations rely on a limited number of time steps (see Figure B2 in the Appendix). In addition, these transformations are shown to lead to models that fit only a limited small range of streamflows. Conversely, some other transformations show a high performance for a large range of streamflows and still lead to a reasonable performance for ex-

treme streamflows; namely, transformations 0.2, *boxcox* and *log* show the best average rank both for the calibration and the evaluation period and may represent adequate transformations to use for many applications. These results should encourage modellers to evaluate the streamflow transformations they use when calibrating hydrological models. The reader may, however, note that complementary aspects may be investigated, such as model robustness when applying hydrological models to climate change applications, peak flows or timing or other models time steps while working on flash flood modelling, for example.

*Code and data availability.* Daily streamflows were retrieved from https://hydro.eaufrance.fr/. The daily SAFRAN atmospheric reanalysis can be retrieved from https://donneespubliques.meteofrance.fr/. The airGR package can be retrieved from https://cran.r-project.org/package=airGR.

### Appendix A:  Impact of the number of intervals used

The whole range of streamflows was split into 200 intervals of equal length. The number of intervals could have an impact on the results. Consequently, we analyse in Fig. A1 how the use of 100 or 500 intervals may impact the conclusions. It appears in this figure that the general shape of all curves remains similar when modifying the number of intervals. However, when 500 intervals are used, the curves are less smooth. This is understandable, since for smaller intervals the results can be more noisy. The main difference stems from the extremes (high and low flows). Indeed, the $-2$ transformation does not reach the top position for low flows and 100 intervals, while it does so for 200 intervals. Correspondingly, the 2 transformation is very close to the top position for 500 intervals and high flows, while it is not so for 100 intervals.

### Appendix B:  Contribution of largest error days of the total error

To assess the contribution of the largest error days of the total error, the concept of fractional contribution to the squared error was used (Newman et al., 2015). The fractional contribution of squared error for the 1, 10, 100, 1000 days with the largest error was calculated for the 11 transformations for GR4J calibrated with NSE over the 325 catchments (Figure B2). It is very clear from this figure that extreme transformations rely on a more limited number of time steps than other transformations. A more complete description of the methodology used to compute the fractional error as well as an example on a single river is shown in Thirel et al. (2023).

### Appendix C:  Maps of catchment characteristics

In this Appendix, the statistics presented in Table 2 are provided as maps.

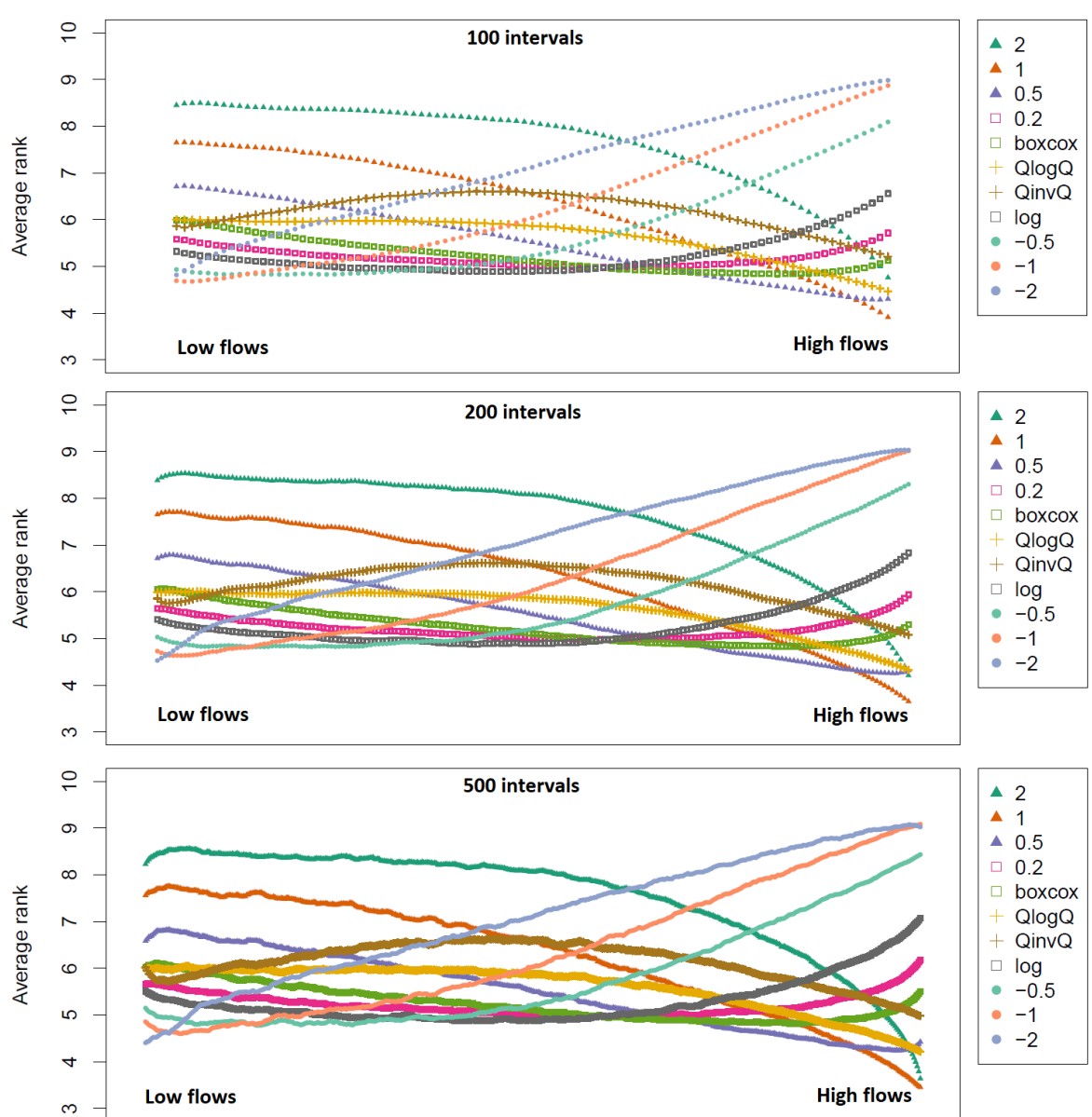

**Figure A1.** Same as Fig. 7 for the GR4J model and the calibration period, for three different numbers of intervals. The model was calibrated with NSE. The $\lambda$ value for the Box–Cox transformation is 0.25.

*Author contributions.* GT and LS conceptualized the study. GT developed the methodology. GT and OD implemented the analyses and visualizations. GT prepared the manuscript with contributions from all co-authors. All authors contributed to the preparation of the revised version of the manuscript.

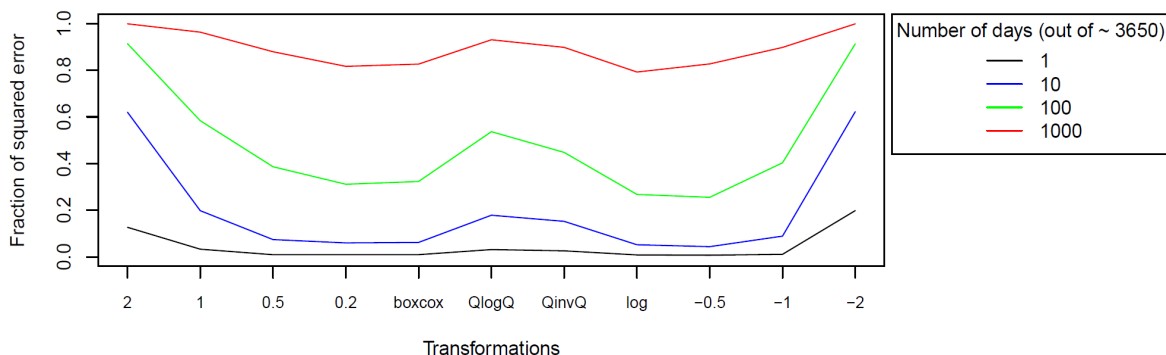

**Figure B1.** Fractional contribution for the GR4J model calibrated with NSE for the Fecht River at Wintzenheim for different numbers of time steps having the highest fractional contribution. Results shown for the calibration period.

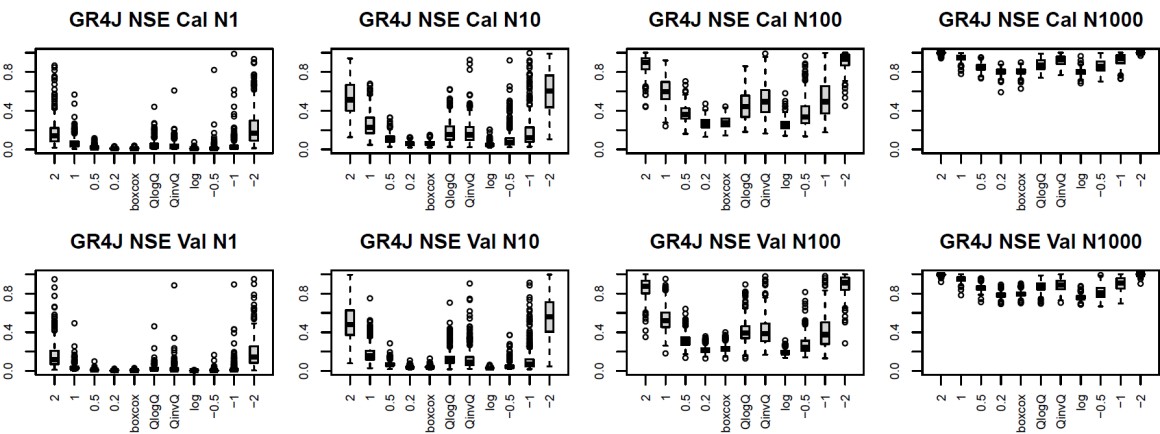

**Figure B2.** Fractional contribution for the GR4J model calibrated with NSE over the 325 stations. N1 to N1000 represent the number of time steps having the highest fractional contribution. Cal means calibration period, and Val means evaluation period. The $\lambda$ value for the Box–Cox transformation is 0.25.

*Competing interests.* The authors declare that they have no conflict of interest.

*Acknowledgements.* SCHAPI and Météo-France are thanked for providing the hydro-meteorological dataset. Laurent Strohmenger and Vazken Andréassian, as well as two anonymous referees, are thanked for their feedback on this study. Isabella Athanassiou is thanked for the manuscript copyediting.

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

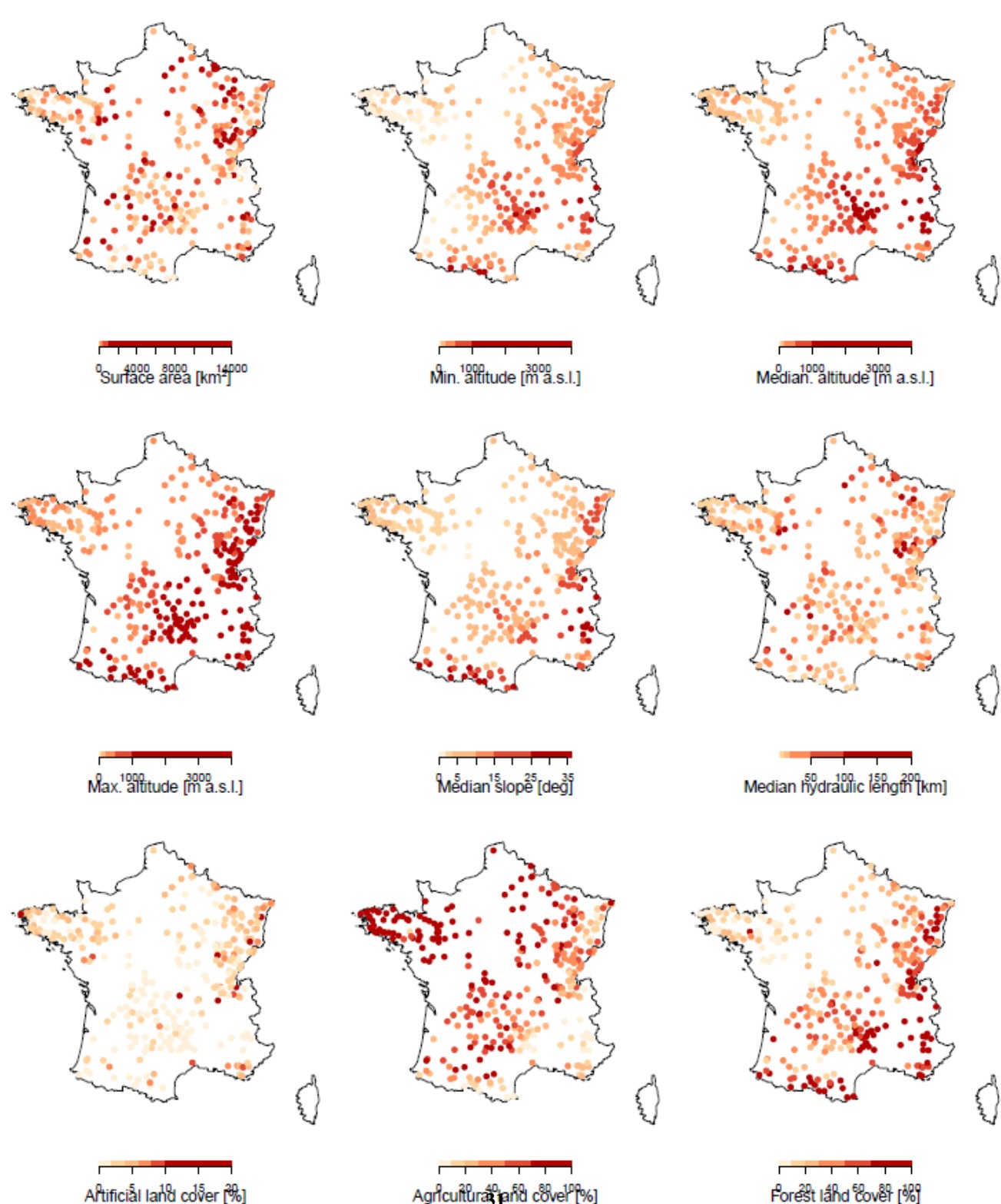

**Figure C1.** Maps of physical characteristics.

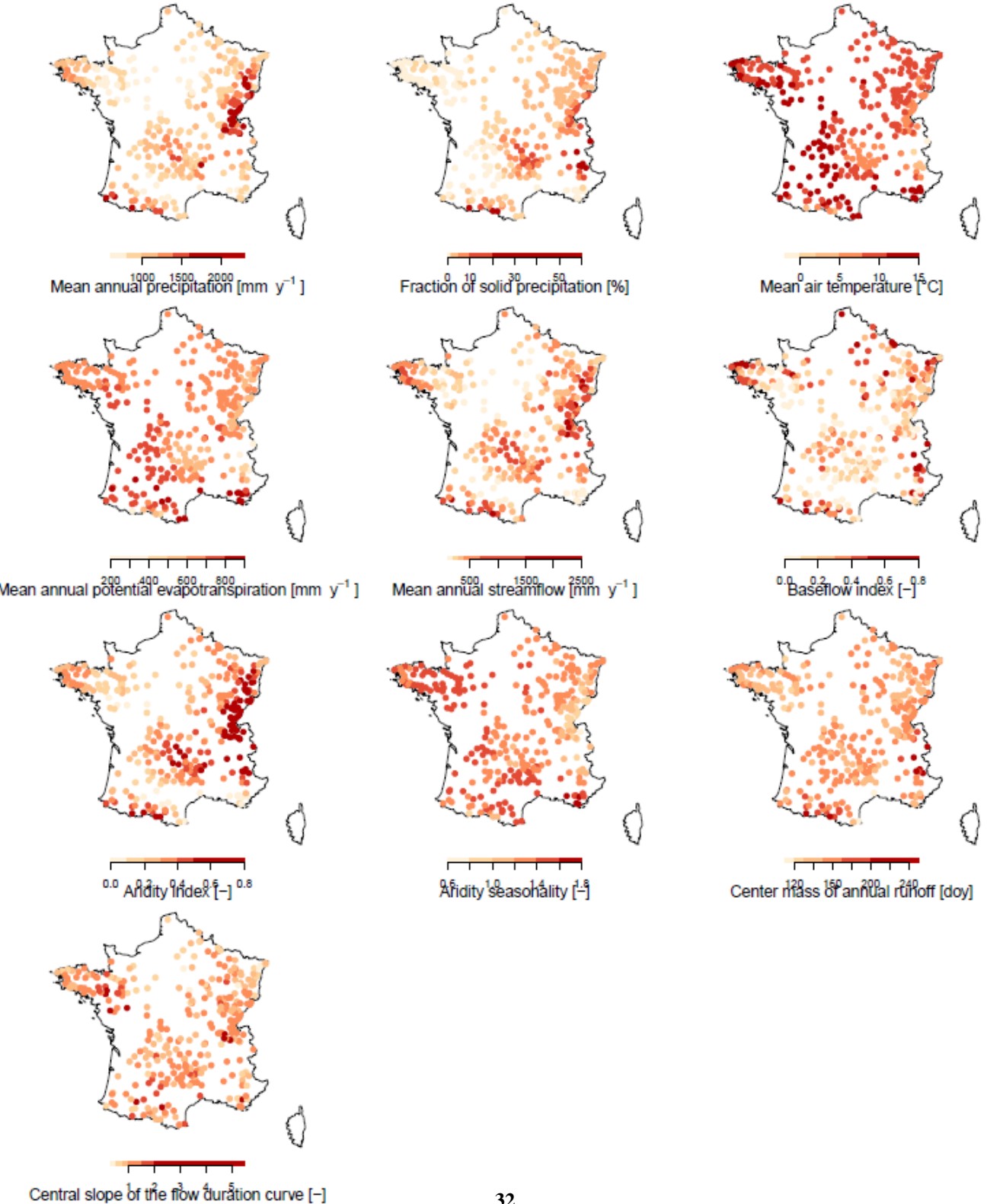

Mean annual precipitation [mm y$^{-1}$]

Fraction of solid precipitation [%]

Mean air temperature [°C]

Mean annual potential evapotranspiration [mm y$^{-1}$]

Mean annual streamflow [mm y$^{-1}$]

Baseflow index [-]

Aridity Index [-]

Aridity seasonality [-]

Center mass of annual runoff [doy]

Central slope of the flow duration curve [-]

**Figure C2.** Maps of indicators over the calibration period.

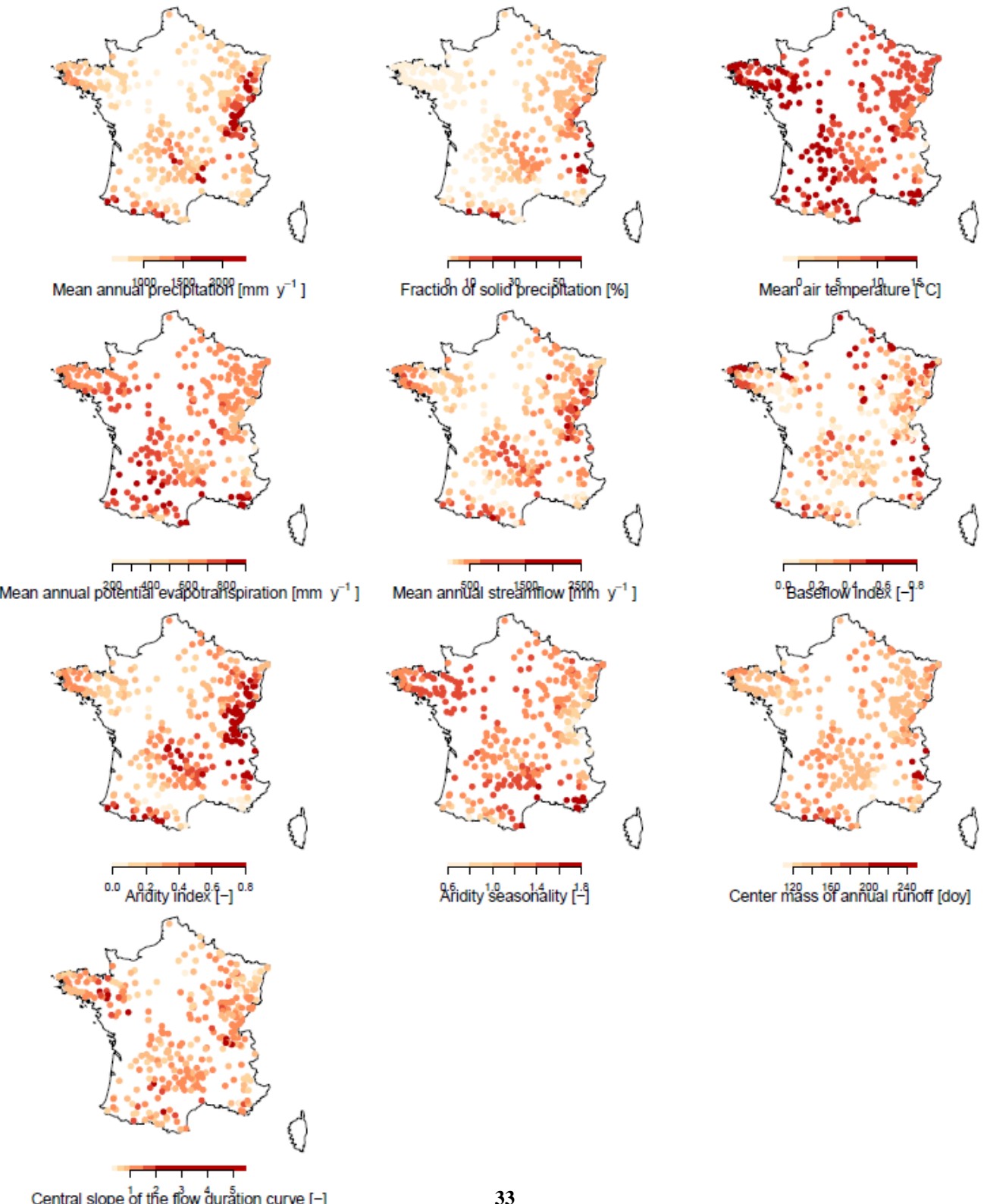

**Figure C3.** Maps of indicators over the evaluation period.