# Peer review of "On the use of streamflow transformations for hydrological model"

_EGUsphere, 2023_

## Author Comment (AC1)

Thank you very much for your review. We provide below some answers to the reviewer's remarks.

[1] In lines 36-46, the authors cite a lot of literature where transformations have been used. I find this paragraph very difficult to read. Would it not be useful to place all these papers in a table and simply report percentages of time a particular transformation has been used? It is quite difficult to find the non-reference text in this paragraph.

A1: We thank the reviewer for this suggestion, which will indeed increase the readability of this paragraph. We will fill in and comment a Table such as the one below:

| Article | $Q^{1/2}$ | $Q^{-1/2}$ | Box-Cox | Other power-law transformations | Inverse |
|---|---|---|---|---|---|
| Smith et al. (2023) | ✓ | ✗ | ✗ | ✓ | ✗ |
| Doe et al. (2023) | ✓ | ✓ | ✓ | ✗ | ✗ |
| … | | | | | |

[2] More explanation would be helpful in places to be clearer about what previous authors found and what the state of knowledge is. The authors cite studies, but it is not clear what relevance the conclusions of these papers have. A couple of examples:

*"Peña-Arancibia et al. (2015) showed that a squared root transformation with the Nash–Sutcliffe efficiency leads to a better calibration and a reduced parameter uncertainty than no transformation or a logarithmic transformation."* – In how far did it lead to better calibration? What does better calibration mean in this context? A better NSE value?

"Sadegh et al. (2018) investigated the role of several transformations in three catchments and two models and deduced that data transformations might be more helpful for evaluation and analysis of model behaviour than model inference." – Why did they conclude that? Why the difference in result for evaluation and inference? Is this conclusion not in conflict with the conclusion of *Peña-Arancibia et al.*? What does 'analysis' mean in this context.

A2: We agree, we will provide more details about these studies.

[3] Why do the authors select these objective functions shown in section 2.3. The authors state that they analyze the following: '*in order to estimate how transformations impact the simulated time series*' . But this is not really what the authors do. They assess performance difference with respect to a couple of popular metrics, they do not analyze how the actual time series changes beyond assessing model performance.

A3: These objective functions were selected for their popularity in hydrological model calibration. In order to assess some sensitivity to the objective function choice, we selected two of them. The piece of sentence '*in order to estimate how transformations impact the simulated time series*' relates to the following: '*the 1995–2005 independent evaluation period is also used*', not to the choice of the objective function. This means that we do not aim to assess only what happens on the calibration period, but also what happens over an independent period after such a calibration, because being applied on periods different from the calibration period is how hydrological models are the most useful.

We respectfully disagree regarding the second part of the reviewer's comment: we do not 'assess the performance difference with respect to a couple of popular metrics'. These metrics are only used as objective functions (combined with transformations). Then, as described in section 3, the performance assessment is purely based on closeness of simulations to observations, i.e. how simulated time series are impacted by the transformations used for calibration.

[4] I am a bit confused by the transformations introduced in section 2.4. Aren't some of the transformations included in others? E.g. the log transformation is a specific case of the Box-Cox transformation. Why not use the minimum number of transformations and then test the influence of the scaling parameter used in the transformation. Using just the Box-Cox transformation and a Q^x transformation with lambda and x varying would capture most and would allow for a more general analysis. You could use the two flexible transformations and plot the result against the lambda and x values used and against the streamflow percentiles to get a better fundamental overview about what is happening!?

A4: We agree on the fact that 'the log transformation is a specific case of the Box-Cox transformation'. However, we decided to stick on the denomination of the actual transformations the most found in the literature. This is a deliberate choice, as we wanted to provide some feedback on transformations that are commonly used. The suggestion of the reviewer would have rather answered to another question, i.e. a try to assess (any?) possible transformations in a more systematic way, in order to finally try to identify a (set of) best transformation(s). Consequently, we prefer to keep the transformations that we selected.

[5] What lambda value has been used for the Box-Cox transformation? The result should be dependent on that choice given that the transformation is flexible. Previous studies suggested a lambda value of 0.3 to suitable for streamflow data to gain a more balanced calibration results (e.g. Vrugt et al. (2006), Journal of Hydrology, doi: 10.1016/j.hydrol.2005.10.041). How much does the result depend on that choice?

A5: We chose a value of 0.25, as suggested by Vazquez et al. (2008) and further used in Santos et al. (2018). We will specify this in the manuscript. The results should indeed depend on that choice but the difference between a lambda value of 0.25 and 0.3 may remain small.

[6] In line 245 you state: *"In addition, the transformations that show the best average rank are not widely used in the literature (0.2, log and boxcox)."*– Are you sure about this? Log and BoxCox (lambda of 0.3) transformations are such a standard to reduce the focus on high flows. They might not have been a focus in very recent years, but certainly from the late 90s to some years ago, they were widely used.

Some (random) examples:

Lerat et al. (2020). Journal of Hydrology, doi.org/10.1016/j.jhydrol.2020.125129

van Werkhoven et al. (2008). Water Resour. Res., doi:10.1029/2007WR006271

Huang et al. (2023). Journal of Hydrology, doi.org/10.1016/j.jhydrol.2023.129347

A6: Thank you for this comment and for the additional references. We agree on the fact that these transformations are used in the literature, we just meant that they are not the most common transformations. We will rephrase as follows "*In addition, the transformations that show the best average rank are not **the most** widely used in the literature (0.2, log and boxcox)."*

[7] For section 4.3, could the authors not organize the catchments into those dominated by slow and fast behavior, e.g. using the (central) slope of the flow duration curve or some other signature metric? There might be different reasons why a catchment varies in this regard (snow, pervious geology, …), which might not be easily captured by the characteristics available.

All in all, an interesting study, though I think the authors could (should?) provide some more fundamental insight still. For example by varying the parameter of the Box-Cox or other flexible transformations.

A7: Thank you for these suggestions. We will add the calculation of the central slope of the flow duration curve, as defined in McMillan et al. (2017). We will also add other indicators such as the ones detailed in answer A25 to the reviewer 2.

Regarding the reviewer's last remark, we agree that assessing in a more systematic way many transformations (e.g. by varying the Box-Cox parameter or by testing many values of power transformations and also potentially by combining criteria) could be interesting. However, we believe that such a study is beyond the scope of the present study, as here we wish to assess transformations selected among those commonly used. We will however add this perspective in the conclusions.

References:

McMillan, H., Westerberg, I., Branger, F.: Five guidelines for selecting hydrological signatures. Hydrol. Process., 31, 4757– 4761, https://doi.org/10.1002/hyp.11300, 2017.

Santos, L., Thirel, G., and Perrin, C.: Technical note: Pitfalls in using log-transformed flows within the KGE criterion, Hydrol. Earth Syst. Sci., 22, 4583–4591, https://doi.org/10.5194/hess-22-4583-2018, 2018.

Vázquez, R. F., Willems, P., and Feyen, J.: Improving the predictions of a MIKE SHE catchment-scale application by using a multi-criteria approach, Hydrol. Process., 22, 2159–2179, https://doi.org/10.1002/hyp.6815, 2008.

---

## Author Comment (AC2)

Thank you very much for your very thorough review. We provide below some answers to the reviewer's remarks.

*Major comments*

1.  Methods: I think that showing the impact of transformations on ranks, rather than on the actual absolute errors that were used to generate those ranks, may hide the real effect that the choice of mathematical functions has on streamflow simulations and, more importantly, may distort a lot the differences in performance (and their perception) among the various types of transformation. For example, what is the difference in mean absolute error between rank 1 and 10? I encourage the authors to show the effects of transformations more directly; for example, they could use a normalized mean absolute error for different streamflow categories, to make the results comparable among catchments.

A1: We thank the reviewer for this interesting comment. The idea behind choosing to work with ranks, instead of direct (normalised) errors was to i) be less impacted by different orders of errors magnitudes between catchments or ranges of streamflows, and to ii) answer the question of what are the best transformations, rather than how good transformations are. We recognize that assigning ranks can have the effect that a rank difference of 1 can both signify a small error or a larger one. We however want to stress out that ranks are accounted for time step by time step, meaning that if differences between two simulations are very low, there can be changes in the order quite easily, which then results in similar average ranks over the intervals.

In order to provide some food for thoughts, we processed as suggested by the reviewer:

1. the hydrological model is calibrated against observed streamflows for a catchment and with a given objective function, successively with different transformations,
2. for each time step, the absolute error is calculated for the simulations obtained with the nine (or 11) transformations,
3. the time series of daily errors are sorted according to the sorted observed streamflow time series,
4. the sorted errors are aggregated over 200 sequential intervals of an equal number of time steps to smooth the results and facilitate the visual analysis.
5. the aggregated sorted errors are normalised by the average of errors over the nine (or 11) transformations, interval by interval.

Applying that to the same example station as in the manuscript, using GR4J calibrated with NSE and results shown over the calibration period, leads to the following plot (which is similar to Figure 6 of the article; please note we use here the new representation as introduced in answer A20 to the reviewer's remarks):

Figure 6 with calculations as suggested by the reviewer:

[Figure]

Actual Figure 6 from the submitted manuscript:

While the general shape of these curves are similar to the one prepared with ranks, it is clear in this new figure that some discrepancies are visible: transformation -2 seems rather far from other transformations most of the time. Another notable difference is that many transformations seem to remain close together for most of streamflow ranges. While this could lead to the conclusion that these transformations can be used interchangeably because they seem to lead to very similar errors, we must note that the very high normalised MSE of one or two transformations leads to smaller differences for the other transformations. In other words, this procedure is impacted by the large error of some transformations and leads to less informative results.

We then automatized this procedure to all 325 stations, with GR4J calibrated with NSE and results shown over the calibration period, and averaged the normalized errors, which gives the following plot (which is similar to Figure 9 of the article):

Figure 9 with calculations as suggested by the reviewer:

[Figure]

Actual Figure 9 from the submitted manuscript:

[Figure]

Here again, we find some similar results as when working with ranks: the general conclusions are not changed. The same groups of transformations still are the best over the same ranges of streamflows.

As a consequence, we prefer not to change the whole methodology used in the manuscript and we will stick to the one proposed so far. We will however mention this other option and discuss it in the revised version.

Additional suggestions to make their analysis more impactful:

- Since NSE is formulated as a function of the sum of squared errors, the authors could report the fractional contribution of the total squared error for the 1, 10, 100, 1000 largest error days obtained with the various transformations (see Figure 10 in Newman et al. 2015). This could provide quantitative support to some statements that the authors make (e.g., L192-193, L338) referring to the number of days where a specific transformation has more weight.

A1.2: We thank the reviewer for this suggestion.

We therefore calculated the fractional contribution to the squared error for the various experiments, to verify if the assertions we made were sound.

First, we want to stress out that the methodology proposed by the reviewer and by Newman et al. (2015) is strictly valid only when we calibrate a model with the NSE objective function, and non-composite transformations. Indeed, NSE and MSE relate to a linear function, as shown by Gupta et al. (2009), in their equation 2:

$$\text{NSE} = 1 - \frac{\sum_{t=1}^{n}(x_{s,t}-x_{o,t})^2}{\sum_{t=1}^{n}(x_{o,t}-\mu_o)^2} = 1 - \frac{\text{MSE}}{\sigma_o^2}$$

Therefore, NSE relates to the squared error (SE) with a linear function too, and the fractional contribution to the SE, for a given time step t1, as written in the following equation, corresponds to the contribution to the NSE:

$$Frac\_contr(t1) = \frac{(Qobs(t1) - Qsim(t1))^2}{\sum_{t=1}^{n}(Qobs(t) - Qsim(t))^2}$$

In addition, for combined transformations QinvQ and QlogQ, we considered the fractional contribution of a time step as the average of the fractional contributions of the two transformations (1 and -1, or 1 and log transformations).

This leads to the Figure below for illustrating our assertion of lines 192-193. This figure gives the fractional contribution of squared error for the 1, 10, 100, 1000 days with the most error, for the 11 transformations for GR4J calibrated with NSE on the Fecht River, and over the calibration period. This illustrated very clearly that for transformations 2 and -2, there is a large weight on very few days for the objective function calculation. For instance, more than 60 % of the contribution rely on 10 days for these transformations, whereas it is lower than 20 % for other transformations. This figure will be added as a supplementary material to justify our assertion.

[Figure]

This analysis was extended to all 325 catchments and is presented in the figure below. In this figure, we still use the GR4J model calibrated on NSE. N1 to N1000 represent the number of time steps having the highest fractional contribution. Cal and Val mean respectively calibration period and validation period. It is here again very clear that extreme transformations rely on a more limited number of time steps than other transformations. This figure will be added as a supplementary material to justify our assertion. Similar results are obtained with GR5J and GR6J, but they will not be shown in the supplementary material.

[Figure]

Finally, as the KGE cannot be written as a linear function of MSE, there is no such straightforward relationship between the term above and the objective function, when the objective function is the KGE or KGE'. Still, we produced these analyses also for KGE and KGE'-based calibrations. As they lead to very similar plots, we chose not to add them in the supplementary material.

- Show the impact on some streamflow characteristics (e.g., Pool et al. 2017), also known as hydrological signatures (e.g., Addor et al. 2018; McMillan 2020).

A1.3: Thank you for your comment. In addition to the already-present mean annual streamflow and baseflow index signatures, we will add the central slope of the flow duration curve as recommended by reviewer 1, but also the aridity index and the center mass of annual runoff (see answer A25).

2. In my opinion, some figures are incredibly complex (e.g., Figures 5 and 8), making the communication of the main messages unnecessarily cumbersome. What do the numbers 1 to 11 represent? Are they related to the number of transformations? Figures 6, 9, 10, 11 and 12 are better to show inter-method differences, though these could (should?) show results of actual mean absolute errors. Additionally, Figure 10, 11 and 12 could be merged into one to facilitate the comparison (the same comment applies to Tables 3, 4 and 5).

A2: We apologize for those complex figures, which we believed would provide additional information to other figures. The numbers 1 to 11 represent the ranks, as written in the y-label and in the caption. They are therefore indeed related to the number of transformations, as the best transformation is ranked first, and the worst is ranked 11[th]. Figures 6, 9, 10, 11 and 12 show the average ranks. It means that they show an aggregated information compared to Figures 5 and 8, which rather show the distribution of ranks. If the

reviewer believes that Figures 6, 9, 10, 11 and 12 are sufficiently informative, we will stick to these ones and remove Figures 5 and 8.

We could merge Figures 10, 11 and 12. However, it means that we would have to reduce their size and therefore their readability would be worse. In addition, and maybe more important, the reader will have to go back and forth in its reading, which could make it uncomfortable. The same goes for the Tables 3, 4 and 5. For these reasons, we would prefer not to merge them.

***Minor comments***

3. L9-10: "…can sometimes be different from what could be expected…". I recommend the authors avoid including vague sentences like this throughout the manuscript, especially in the abstract.

A3: We propose the following: '… can sometimes be different from their expected behaviour".

4. L19-20: From my view, there is general consensus in the community that no universal hydrological model structure exists, since each one is an assembly of hypotheses on the functioning of a specific hydrological system (Clark et al. 2011). This has motivated a proliferation of flexible modeling platforms such as FUSE (Clark et al. 2008), SUPERFLEX (Fenicia et al. 2011), Noah-MP (Niu et al. 2011), SUMMA (Clark et al. 2015a,b, 2021), MARRMoT (Knoben et al. 2019), Raven (Craig et al. 2020) and even airGR with its variants GR5J and GR6J. I think this is a good place to make this point.

A4: We do agree about this consensus and this is what we tried to express here. We will rephrase with the reviewer's proposition 'there is consensus in the community that no universal hydrological model structure exists' and we will add the references the reviewer provides.

5. L24: This is a good place to cite previous studies showing the impact of subjective calibration criteria selection on hydrological modeling applications (e.g., Mendoza et al. 2016; Fowler et al. 2018; Melsen et al. 2019).

A5: Thank you for these suggestions, we will cite these references.

6. L33: I think you should refer to Figure 1a.

A6: We do not understand this comment, as we wrote 'This is illustrated in Fig. 1, where in panel a, the larger errors'.

7. L52-58: I suggest citing these studies in chronological order.

A7: We will modify the order of sentences to cite these studies in a chronological order.

8. Figure 1: I suggest including the model being used and the simulation year in the figure caption.

A8: We did not specify the model and simulation year as we believed that this information was not useful here, as what we wanted to show was not actual events, but the behaviour of streamflow transformations. However, we will add it in the revised version of the manuscript.

9. L70-72: This sentence is very confusing. "Alteration" may be interpreted by some readers as human intervention. I suggest re-wording.

A9: Thank you. We propose "with specific streamflow selection procedures such as…".

10. L82-83: Did the authors examine whether the calibration and evaluation periods are hydroclimatically different? Please clarify.

A10: We wrote in lines 87-88, "This table also shows that the climatic conditions are similar between the two periods, with the evaluation period being only slightly warmer and wetter than the calibration period ». To further investigate potential hydroclimatic differences between the two periods, we propose below boxplots showing the difference of annual precipitation (left), air temperature (middle) and discharge (right) between the two periods. The boxplots are composed of 325 values, i.e. one for each catchment. These boxplots confirm our assertion as the differences for the median are limited for precipitation and air temperature and very low for discharge. Some catchments show larger differences, but those are in a limited number. The boxplots are shown below but will not be included in the manuscript.

[Figure]

11. I think that much of the text in section 2.2 corresponds to methodology, and therefore should be included in the methods section.

A11: We understand the reviewer's concern. We propose some renaming of sections and subsections, as follows:

2. Material and method

2.1. Catchment set and data

2.2. Hydrological model

2.3. Optimization criteria

2.4. Streamflow transformations

2.5. Evaluation methodology (former section 3 Methods)

Then section 4 Results becomes section 3 Results.

12. L98: Can you please clarify how you determined snowfall occurrence in your basins?

A12: We made the approximation that if the daily temperature is below 0 °C, then this is snowfall (as in Knoben et al., 2018). We will clarify this in the manuscript.

13. L101: Why did you choose five elevation bands and not more/less? Did you try other configurations? I think this needs a proper justification, given the large effects that this decision may have on simulated states and fluxes (Murillo et al. 2022).

A13: This choice was based on the previous works led by our colleagues in the past, who assessed the added value of using different numbers of elevation bands and concluded that this number represents a good compromise between time calculation, model efficiency and data quality (see Valéry et al., 2014 and Valéry, 2010). Since the focus of the article is not on

this issue, we think that the cited reference provides sufficient information to the reader to justify this choice.

14. Table 1: I think it would be more informative to show these attributes as maps with a color bar (see, for example, Addor et al. 2017; Alvarez-Garreton et al. 2018).

A14: We made some tests to investigate whether it was possible to replace this table with a figure as requested by the reviewer. We show in the answer A25 these maps. In order to keep the number of figures limited, as requested by the reviewer, and because we believe that maps are less readable than the table, these figures will be inserted as a supplementary material.

15. L111: please specify whether your simulations consider a spin-up period.

A15: We definitely used a spin-up period. A 1-year period, corresponding to the year preceding the calibration or the evaluation period, was used. We will add this information.

16. L160-163: I think this text should be in the methods section.

A16: We saw this text as an introduction about how the results will be presented in the following. As the reviewer thinks that this is clearer to put this information in the Methods section, we will move it there.

17. Figure 3d: the numbers in the y axis are not legible.

A17: Sorry for that, the numbers were somehow cut during the production process, we will correct that.

18. Figure 5 (caption): is CemaNeige implemented in this basin?

A18: No, we did not use CemaNeige here. We will specify it in the caption.

19. L185: 'average rank of transformations'. How do you compute that average?

A19: Each value of each curve is simply the average of the ranks of transformations over all the time steps of the concerned interval. We will specify it in the text.

20. L185-190: all these comparisons are very hard to visualize. You could use symbols in Figures 6 and 9 to 12 to help readers to see what you want them to see. For example, use X for negative transformations, square for log, circle for Box-Cox, etc.

A20: Thank you for this suggestion. We propose the following visualization, here replacing Fig. 6 of the manuscript. All similar figures will be replaced with this visualization.

[Figure]

A21: We mean that over the calibration periods, the transformations can lead to simulations that are relatively good for their supposed target (e.g. transformation -2 has a low average rank over low flows), but since this average rank is higher for the evaluation period, we can consider that the transformations are less specific, i.e. worse for their supposed target, and closer to each other. We will rephrase and we propose the following:

'To phrase it differently, over the evaluation period, the transformations lead to simulations that are less specific, i.e. closer to each other.'

22. L279: 'to behave much worse'. Note that you are judging based on the ranks, and not on the actual sum of absolute errors. I think it would be much more honest if you showed the latter.

A22: We understand the concern of the reviewer. We will rephrase this sentence to be fairer with what is actually shown. We propose the following: "which appears to show much worse ranks than...".

**23. L296: You have ranks for 9/11 transformations. Did you obtain the same number of correlations?**

A23: In this section, we discuss results for the GR4J model calibrated with NSE, which means that we have 11 ranks. Correlations were also calculated for GR4J calibrated with KGE, resulting indeed into 9 correlation values, but we did not discuss these results as no further informative result arose.

**24. L301: Are these correlations statistically significant?**

A24: We thank the reviewer for this suggestion.

First, we must mention that we calculated the correlations for the new characteristics suggested by the reviewers (see the Table provided in A25). Unfortunately, no outstanding correlations were found. We observed that the central slope of the flow duration curve shows correlations of the same order of magnitude, but opposite to the one shown by the BFI, with exactly the same transformations. This indicates that these indicators are somehow well related.

Second, the p-values were calculated for all the calculated correlations using the stats::cor.test() function in R. We found that the correlations lower than -0.11 and higher than 0.11 were all significant (i.e. p-values < 0.05), whereas none of the other ones were significant. Consequently, this does not change the related analysis.
We will add this information in the revised manuscript.

**25. Section 4.3: I suggest the authors adding to their analysis the aridity index, the seasonality of aridity (Knoben et al. 2018) and maybe the center of time of runoff (Stewart et al. 2005).**

A25: We calculated these indicators as suggested (in addition to flow signatures already mentioned) and added them to our analysis and to Table 1. We assume that the "center of time of runoff" mentioned by the reviewer corresponds to the "timing of the center of mass of the annual runoff" as introduced by Stewart et al. (2005).

Table 1 will therefore be modified as shown below. We can see that for the four new indicators the catchments show some variability between each other, but also that the two periods seem to face rather similar conditions, as was already the case for other indicators. This will be mentioned in the paper.

| Characteristic | Period | Minimum | Median | Maximum |
|---|---|---|---|---|
| Surface area [km$^2$] | - | 5.3 | 225.5 | 13 483.5 |
| Min. altitude [m a.s.l.] | - | 6.0 | 209.0 | 2 154.0 |
| Median altitude [m a.s.l.] | - | 53.0 | 368.0 | 2 741.0 |
| Max. altitude [m a.s.l.] | - | 93.0 | 784.0 | 3 997.0 |
| Median slope [deg] | - | 1.1 | 7.4 | 35.8 |
| Median hydraulic length [km] | - | 2.1 | 19.0 | 200.7 |
| Artificial land cover [%] | - | 0.0 | 2.1 | 18.2 |
| Agricultural land cover [%] | - | 0.0 | 54.2 | 97.7 |
| Forest land cover [%] | - | 0.0 | 43.5 | 100.0 |
| Mean annual precipitation [mm y$^{-1}$] | Calibration | 651 | 1 009 | 2 204 |
| | Evaluation | 691 | 1 025 | 2 077 |
| Fraction of solid precipitation [%] | Calibration | 0.3 | 2.5 | 59.1 |
| | Evaluation | 0.0 | 2.2 | 50.3 |
| Mean air temperature [°C] | Calibration | -1.1 | 10.0 | 13.9 |
| | Evaluation | -0.9 | 10.3 | 14.2 |
| Mean annual potential evapotranspiration [mm y$^{-1}$] | Calibration | 252 | 661 | 858 |
| | Evaluation | 267 | 678 | 871 |
| Mean annual streamflow [mm y$^{-1}$] | Calibration | 101 | 405 | 2485 |
| | Evaluation | 123 | 410 | 2250 |
| Baseflow index [−] | Calibration | 0.01 | 0.22 | 0.68 |
| | Evaluation | 0.01 | 0.23 | 0.76 |
| Aridity index [−] | Calibration | 0.03 | 0.33 | 0.74 |
| | Evaluation | 0.01 | 0.33 | 0.77 |
| Aridity seasonality [−] | Calibration | 0.69 | 1.33 | 1.64 |
| | Evaluation | 0.62 | 1.36 | 1.72 |
| Center mass of annual runoff [doy] | Calibration | 117 | 152 | 248 |
| | Evaluation | 113 | 145 | 244 |
| Central slope of the flow duration curve [−] | Calibration | 0.39 | 1.05 | 5.05 |
| | Evaluation | 0.40 | 1.01 | 5.26 |

Table 1: Same as in the article, with the addition of the suggested characteristics

In addition, these characteristics will be represented as maps, and provided in the supplementary material.

[Figure]

Fig: Maps of physical characteristics

[Figure]

Fig: Maps of indicators on the calibration period

[Figure]

Fig: Maps of indicators on the evaluation period

***Some suggested edits***

26. L30: 'have been' -> 'has been' ('a wide panel' is singular).

27. L36: I suggest deleting 'more specifically'.

28. L44: 'some other works' -> 'other studies'.

29. L48-49: delete 'Nevertheless, some authors tried to investigate this issue. For instance,'.

30. L59: 'Still, most of the time' -> 'To the best of our knowledge'.

31. L59: 'are not' -> 'have not been'.

32. L61-62: I strongly encourage the authors to write that finding with their own words instead of quoting.

33. L63 and anywhere else: I recommend the authors using past tense (i.e., 'used' and 'justified') when referring to previous studies.

34. L68: 'tends to illustrate' -> 'illustrates these assertions to some degree'. Delete 'we feel that'.

35. L69: delete 'in this article'.

36. L75: 'Data' -> 'We used data from…'. I strongly motivate the authors to use active voice.

37. L95: 'Maximal' -> 'Maximum'.

38. L101: 'take into account the catchment heterogeneity' -> 'consider intra-catchment variability'.

39. L103: delete 'while GR4J is the main model used' and write 'In this work, we also use the GR6J model to assess the transferability...'.

40. L124: 'with N the total number' -> 'being N the total number'.

41. L131: 'as this focuses' -> 'as it focuses'.

42. L300: Delete 'Unfortunately, only a few correlations could be identified'.

43. L301: 'Anti-correlations' reads really awkward. I suggest writing 'negative correlations' instead.

A26-43: we will consider all these edits during the revision phase, thank you for suggesting them.

**References**

Addor, N., A. J. Newman, N. Mizukami, and M. P. Clark, 2017: The CAMELS data set: Catchment attributes and meteorology for large-sample studies. *Hydrol. Earth Syst. Sci.*, doi:10.5194/hess-21-5293-2017.

Addor, N., G. Nearing, C. Prieto, A. J. Newman, N. Le Vine, and M. P. Clark, 2018: A Ranking of Hydrological Signatures Based on Their Predictability in Space. *Water Resour. Res.*, **54**, 8792–8812, doi:10.1029/2018WR022606.

Alvarez-Garreton, C., and Coauthors, 2018: The CAMELS-CL dataset: Catchment attributes and meteorology for large sample studies-Chile dataset. *Hydrol. Earth Syst. Sci.*, **22**, 5817–5846, doi:10.5194/hess-22-5817-2018.

Clark, M. P., A. G. Slater, D. E. Rupp, R. A. Woods, J. A. Vrugt, H. V. Gupta, T. Wagener, and L. E. Hay, 2008: Framework for Understanding Structural Errors (FUSE): A modular framework to diagnose differences between hydrological models. *Water Resour. Res.*, **44**, W00B02, doi:10.1029/2007WR006735.

——, D. Kavetski, and F. Fenicia, 2011: Pursuing the method of multiple working hypotheses for hydrological modeling. *Water Resour. Res.*, **47**, W09301, doi:10.1029/2010WR009827.

Clark, M. P., and Coauthors, 2015a: A unified approach for process-based hydrologic modeling: 1. Modeling concept. *Water Resour. Res.*, doi:10.1002/2015WR017198.

——, and Coauthors, 2015b: A unified approach for process-based hydrologic modeling: 2. Model implementation and case studies. *Water Resour. Res.*, doi:10.1002/2015WR017200.

Clark, M. P., and Coauthors, 2021: The numerical implementation of land models: Problem formulation and laugh tests. *J. Hydrometeorol.*, **22**, 1627–1648, doi:10.1175/JHM-D-20-0175.1.

Craig, J. R., and Coauthors, 2020: Flexible watershed simulation with the Raven hydrological modelling framework. *Environ. Model. Softw.*, **129**, 104728, doi:10.1016/j.envsoft.2020.104728. https://doi.org/10.1016/j.envsoft.2020.104728.

Fenicia, F., D. Kavetski, and H. H. G. Savenije, 2011: Elements of a flexible approach for conceptual hydrological modeling: 1. Motivation and theoretical development. *Water Resour. Res.*, **47**, W11510, doi:10.1029/2010WR010174.

Fowler, K., M. Peel, A. Western, and L. Zhang, 2018: Improved Rainfall-Runoff Calibration for Drying Climate: Choice of Objective Function. *Water Resour. Res.*, **54**, 3392–3408, doi:10.1029/2017WR022466.

Knoben, W. J. M., R. A. Woods, and J. E. Freer, 2018: A Quantitative Hydrological Climate Classification Evaluated With Independent Streamflow Data. *Water Resour. Res.*, **54**, 5088–5109, doi:10.1029/2018WR022913. https://onlinelibrary.wiley.com/doi/abs/10.1029/2018WR022913.

——, J. E. Freer, K. J. A. Fowler, M. C. Peel, and R. A. Woods, 2019: Modular Assessment of Rainfall–Runoff Models Toolbox (MARRMoT) v1.2: an open-source, extendable framework providing implementations of 46 conceptual hydrologic models as continuous state-space formulations. *Geosci. Model Dev.*, **12**, 2463–2480, doi:10.5194/gmd-12-2463-2019.

McMillan, H., 2020: Linking hydrologic signatures to hydrologic processes: A review. *Hydrol. Process.*, **34**, 1393–1409, doi:10.1002/hyp.13632.

Melsen, L., A. J. Teuling, P. J. J. F. Torfs, M. Zappa, N. Mizukami, P. A. Mendoza, M. P. Clark, and R. Uijlenhoet, 2019: Subjective modeling decisions can significantly impact the simulation of flood and drought events. *J. Hydrol.*, **568**, 1093–1104, doi:10.1016/j.jhydrol.2018.11.046.

Mendoza, P. A., M. P. Clark, N. Mizukami, E. D. Gutmann, J. R. Arnold, L. D. Brekke, and B. Rajagopalan, 2016: How do hydrologic modeling decisions affect the portrayal of climate change impacts? *Hydrol. Process.*, **30**, 1071–1095, doi:10.1002/hyp.10684.

Murillo, O., P. A. Mendoza, N. Vásquez, N. Mizukami, and Á. Ayala, 2022: Impacts of Subgrid Temperature Distribution Along Elevation Bands in Snowpack Modeling: Insights From a Suite of Andean Catchments. *Water Resour. Res.*, **58**, under review, doi:10.1029/2022WR032113.

Newman, A. J., and Coauthors, 2015: Development of a large-sample watershed-scale hydrometeorological data set for the contiguous USA: data set characteristics and assessment of regional variability in hydrologic model performance. *Hydrol. Earth Syst. Sci.*, **19**, 209–223, doi:10.5194/hess-19-209-2015. http://www.hydrol-earth-syst-sci.net/19/209/2015/.

Niu, G.-Y., and Coauthors, 2011: The community Noah land surface model with multiparameterization options (Noah-MP): 1. Model description and evaluation with local-scale measurements. *J. Geophys. Res.*, **116**, D12109, doi:10.1029/2010JD015139.

Perrin, C., C. Michel, and V. Andréassian, 2003: Improvement of a parsimonious model for streamflow simulation. *J. Hydrol.*, **279**, 275–289, doi:10.1016/S0022-1694(03)00225-7.

Pool, S., M. J. P. Vis, R. R. Knight, and J. Seibert, 2017: Streamflow characteristics from modeled runoff time series - Importance of calibration criteria selection. *Hydrol. Earth Syst. Sci.*, **21**, 5443–5457, doi:10.5194/hess-21-5443-2017.

Pushpalatha, R., C. Perrin, N. Le Moine, T. Mathevet, and V. Andréassian, 2011: A downward structural sensitivity analysis of hydrological models to improve low-flow simulation. *J. Hydrol.*, **411**, 66–76, doi:10.1016/j.jhydrol.2011.09.034. http://dx.doi.org/10.1016/j.jhydrol.2011.09.034.

Stewart, I. T., D. R. Cayan, and M. D. Dettinger, 2005: Changes toward earlier streamflow timing across western North America. *J. Clim.*, **18**, 1136–1155, doi:10.1175/JCLI3321.1.

Valéry, A., V. Andréassian, and C. Perrin, 2014: 'As simple as possible but not simpler': What is useful in a temperature-based snow-accounting routine? Part 2 – Sensitivity analysis of the Cemaneige snow accounting routine on 380 catchments. *J. Hydrol.*, **517**, 1176–1187, doi:https://doi.org/10.1016/j.jhydrol.2014.04.058.

References:

Valéry, A., 2010. Modélisation précipitations – débit sous influence nivale. Élaboration d'un module neige et évaluation sur 380 bassins versants. Thèse de Doctorat, Cemagref (Antony), AgroParisTech (Paris), 405 pp.

Valéry, A., V. Andréassian, and C. Perrin, 2014: 'As simple as possible but not simpler': What is useful in a temperature-based snow-accounting routine? Part 2 – Sensitivity analysis of the Cemaneige snow accounting routine on 380 catchments. *J. Hydrol.*, **517**, 1176–1187, https://doi.org/10.1016/j.jhydrol.2014.04.058.

---

## Author Response (AR1)

We would like to thank again the two reviewers for the work on reviewing the manuscript, which greatly helped us to improve it. We provided detailed answers to the reviews during the open discussion phase. All the modifications we made in the manuscript correspond to what we said we would do and we neither made further modifications nor turned back on the propositions we made.

---

## Author Response (AR2)

*Reviewers' comments are in italics.* Our responses are in blue.

Reviewer #1

*The authors mainly addressed my comments. Some not as extensively as I would have thought to be useful, but there are.*

*A reamining point I find very unhelpful is that the authors refer to the BoxCox transformation as such in all tables and figures. However, this is based on a single choice of lambda value. This should be highlighted in tables/figures, e.g. include the chosen value in brackets).*
*It would also be appropriate for the authors to at least briefly describe in the text that different choices for lambda have been used in the hydrologic literature. I had given suggestions for references in my previous review.*

We thank the reviewer for this suggestion. We added the value used for lambda in all captions of the tables and figures where the Box-Cox transformation was used in our work. In addition, we specified the value used for lambda in table 1, presenting a non-exhaustive list of references using streamflow transformations. We also added in table 1 the references the reviewer suggested in his/her previous review. We apologize for this omission.

Reviewer #2

*I would like to thank the authors for considering my suggestions. After the first round of reviews, the manuscript is technically much stronger, and the collection of figures and tables supports most of the conclusions. However, I think that the writing needs to be improved to maintain the high standards of HESS. Additionally, one conclusion is unsupported, and the manuscript lacks any discussions on how the results connect with the literature cited in the introduction.*

We thank the reviewer for their general assessment of the manuscript, and for their very detailed reading and suggestions, which will improve the quality of the manuscript.

*Major comments*

*1. The number of awkward/redundant sentences is so large that interrupts the flow of the reading in nearly all sections. I think that the authors need to make an important effort in revising the text before this manuscript can be considered for publication. Given all the available tools and resources (e.g., Google translator, Grammarly), a final version written in good English should be easy to achieve.*

We thank the reviewer for these suggestions. We had already asked a professional translator to copyedit the manuscript before the initial submission. The revision process may have introduced incorrect style or grammar formulations. We asked for a new correction before resubmission and hope that this will help to meet the standards of the journal.

*Some examples:*
*- L2-3: "It is a widespread technique that has been…". This sentence is completely redundant and*

*should be deleted.*
*- L3: "Indeed" -> "Further".*
*- L4-5: "Besides, the actual goal of the model application… is undertaken". This sentence is redundant and distracting. Delete.*
*- L8-L9: "Typically, a logarithmic transformation…no transformation". This sentence is out of place here.*
*- L14: "intermediate range" -> "medium range".*
*- L19: "impact" -> "impacts".*
*- L23: "Consequently" and "such as those mentioned above" are completely redundant and should be deleted.*
*- L25: "on the use of a criterion (sometimes a combination of criteria)" -> "on the use of one or more criteria".*
*- L28 :"and two different chosen criteria" -> "and two different criteria".*
*- L32: "wide panel" -> "wide range".*
*- L33: "…has been introduced in the literature (Bennett et al., 2013). These transformations consist…" -> "…has been introduced in the literature (Bennett et al., 2013), which consist…"*
*- L46-47: "in a review of suitability…. of low flows" -> redundant and distracting. I suggest deleting.*
*- L47: "justify" -> "justified". You should use past tense when referring to what was done in the past.*
*- L57: "To the best of our knowledge, the use and choice of transformation have not been thoroughly assessed". Because you quote some studies afterwards, it would be more appropriate to write "Only a few studies have assessed…" (or something similar).*
*- L52: delete "calibration that provides".*
*- L53: delete "leads to".*
*- L75: "possibly identify" -> "explore possible links".*
*- L80-81: I suggest writing "Daily meteorological and hydrological data from the period 1985-2005 were used…".*
*- L83: "…in Table 2. It illustrates the high diversity" -> "…in Table 2, illustrating the large diversity…".*
*- In the caption of Table 2, I suggest removing "statistics of the" from the beginning. Replace the last sentence by "The maps with sample statistics for these catchment features are included in Appendix C".*
*- L129: I suggest re-writing as "The hydrological models are calibrated by applying different transformations to streamflow values in the calculation of the objective functions" (or something like that).*
*- L139-140: I suggest rewriting as "In order to evaluate the impact of the transformations on model calibration, we use a common analysis framework that aims at…".*
*- Caption of Figure 3: no need for capital letter after ";" in "…series; b) Absolute…".*
*- L166: rewrite as "Figure 4 illustrates an example application for a single catchment…".*
*- L190: delete "quite logically". Let the readers judge on this.*
*- L209: delete "some trends can be observed". Or replace by "some general features can be observed" (the word "trend" is typically associated with "temporal trends").*
*- L223: replace "This result is interesting since most of the time" by "Typically".*
*- L237-238: "However, the purpose of using models is to apply them under conditions that are different from those they are calibrated on."*
*- L286-287: awkward sentence. Please re-word.*
*- L313: "on the basis of" -> "using".*
*- L346: "flood peak" -> "peak flows".*
*- Appendix A and B: I suggest deleting "On the" from the title.*
*- L359: "the most error" -> "the largest error".*

All these modifications were made. The reviewer's suggestions that were not strictly followed are detailed below.

*- L28: "will impact differently on the calibration process" -> "will impact differently the calibration process".*

We rephrased as follows: "will impact the calibration process differently".

*- L32: "criteria" -> do you mean objective functions? It would be good idea to be more specific.*
Here we actually wanted to be rather general. This sentence is true for objective functions, but also when just assessing the quality of a model simulation, i.e., for a performance criterion.

*- L39: "Since many metrics are squared metrics" reads weird. You could replace by "Since many metrics rely on squared transformations…".*

We agree. However, we do not want to use the word « transformation » here, as the square operator is not applied to simulation and observation time series, but rather to the difference between these time series. We suggest "Since many metrics rely on squared errors…".

*- L42-43: since you have a summary table, you could replace this sentence with something like "A non-exhaustive list of transformations is listed in Table 1, being XX and YY the most popular".*

We rephrased as "A non-exhaustive list of transformations is listed in Table 1, with the square root, the logarithmic, the reciprocal of squared root, the inverse or other power–law transformations being the most popular".

*- L58-66: I find the number of quotes from past papers excessive. I motivate the authors to describe previous findings that are relevant for their research using their own words.*

We understand that the quotes may appear excessive here. However, we wanted to remain as close as possible to the authors' explanations, not to distort the points of view they express. Therefore, we would prefer to stick to the current formulation.

*- L237-238: "However, the purpose of using models is to apply them under conditions that are different from those they are calibrated on."*

We did not understand the reviewer's suggestion, since it corresponds to the original sentence. If the reviewer found it unclear, we propose rephrasing it as: "However, the purpose of using models is to apply them on periods different from those used for calibration" (see our answer to comment 16).

*2. L268-269: "We can therefore conclude that the analysis is only slightly dependent on the model used" (also in L333-334). I think that the experimental setup does not enable to conclude this, unless*

*they repeat their calibration experiments using model structures with very different degrees of process and spatial complexity (sampling, for example, the model space described by Hrachowitz and Clark 2017 in Figure 1). Despite this is a major concern, it can be easily addressed by replacing that sentence by "For the models used here, the relative performance of transformations is very similar across the streamflow range", and re-wording similar statements in the conclusions section.*

We thank the reviewer for this remark. We agree that models with conceptual differences much more important than those between GR4J and GR6J could lead to additional understanding of the actual contribution of model structure on such a conclusion. While we think that models with different spatial complexities would open another research question, lumped models with different conceptualization exist. Although we believe that the end of the sentence pointed out by the reviewer conveys a similar message as the reviewer's remark (", although we must bear in mind that these models are partially similar"), we modified the sentence as suggested by the reviewer, and reworded the conclusions section.

*3. I think that the authors should make an effort to connect their results with the existing literature. This can be done in a separate section named "Discussion", or within the results section, renaming it to "Results and Discussion".*

Thank you for this comment. We agree that connecting our results with the existing literature could provide added value to the manuscript. Due to the limited literature about this specific research question addressed in this work, we opted to add a subsection in the results section, renaming it Results and Discussion as suggested by the reviewer.

We also modified the following sentence in the conclusions: "They show that no a priori assumption on streamflow transformations can be taken as warranted." as follows : "They show that, although some common beliefs about the impact of transformations are confirmed by this study, no a priori assumption on streamflow transformations can be taken as warranted."

*Minor comments*

*4. L67-71: I think this is a good place to clearly describe the gap(s) that this study intents to fill or, even better, state the research question(s) justifying the existence of this paper.*

The reviewer is right. We attempted to improve this paragraph accordingly.

*5. L80: I suggest moving the link to the data availability statement.*

Done.

*6. Table 2: I don't think you need decimals for altitude. I suggest replacing "mean annual streamflow" by "mean annual runoff", since your units are not volume per time units.*

Done.

*7. L98-99: Did you check whether excluding CemaNeige affects your results for these catchments?*

Not using CemaNeige strongly affects the hydrological simulations when the proportion of snow becomes high, as the hydrological regime cannot be well reproduced. We believe that the missing process (snow accumulation and melt) could be wrongly compensated by transformations and the calibration algorithm. Consequently, comparing transformations on snowfed catchments without using CemaNeige was not checked.

*8. I suggest renaming section 2.3 to "Calibration metrics" or "Objective functions", since the optimization criteria also involves the choice of optimization algorithm (which was already described).*

Done.

*9. L148: it would be good to clarify that in Figure 3 you only have 1 or 2 because you have only two transformations (right?).*

The reviewer is right. Done.

*10. Figure 3: the numbers in the y-axis are too small.*

We increased the size of the fonts.

*11. Figure 4: Please increase the size of characters, especially for the transformations on the right.*

We increased the size of the fonts.

*12. L194-195: "To circumvent this issue…". This sentence should be in the methods section.*

We respectfully disagree with this recommendation. The fact that we worked one a single catchment, and then on 325 catchments to generalize the results, is already explained at the end of the methods section. Here, we want to recall this fact, and to further justify it with the results obtained on a single catchment (which were by definition not available in the methods section). We slightly modified the sentence as follows: "To circumvent this issue, and to generalize the results, we perform a similar analysis over the 325-catchment set presented in section 2."

*13. L197-198: do you mean the most number 1 ranks among the 325 catchments?*

Yes.

*14. L271: I do not see an arc shaped curve for QlogQ. Please revise and correct if needed.*

The QlogQ transformation does not appear in this Figure (Figure 10), as the use of log transformations is not advised for KGE (see cited reference in the text).

*15. L219-220: I do not see what the authors write for the boxcox transformation. The minimum is reached close to the high flow category.*

The reviewer is right. We modified as follows: « Finally, transformation *boxcox* shows the best rank for medium to high flows, but not for the highest flows. »

*16. L237: The authors state that "the purpose of using models is to apply them under conditions that are different from those they are calibrated on". I think that having this sentence here is misleading, since the hydroclimatic conditions defined in this study for the calibration and evaluation periods are very similar (L86-87). I suggest deleting or re-wording.*

We partially agree, as we used an independent period that is by definition different from the calibration period. However, it is true that we did not seek a climatologically different period. We suggest the following reformulation: « However, the purpose of using models is to apply them on periods different from those used for calibration. »

*17. L255-259: this text should be in the methods section.*
*18. L277-282: this text should be in the methods section.*

The tests linked to this text were already mentioned in the methods section. However, recalling the additional tests in the results helps the reader to understand the rationale of performing them based on previous findings. We therefore prefer to keep them. We also added the experiment plan at the end of the methods section.

*19. L283-284: Does this mean that catchments with low BFI and calibrated with QlogQ will have low ranks (i.e., poorer performance) compared to other transformations? I recommend the authors explaining here the practical implications of their results.*

Thank you for this comment. We could indeed help the readers to understand the implications of this analysis. For this example, the reviewer is almost right: it means that catchments with low BFI and calibrated with QlogQ will have low rank values, and thus a higher performance (rank 1 being the lowest error). We added this sentence: "It means that the lower the BFI, the better the use of transformations giving intermediate weight between high and low flows. Conversely, the higher the BFI, the better the use of transformations that give a large weight to low flows."

*20. Figure 10: I recommend to adjust the limits in the y-axis (e.g., by setting maximum values to 7) for comparison purposes.*

We decided to keep for Figure 10 the same limits for the y-axis as for Figure 9 and previous figures, for the sake of homogeneity with previous figures.

*21. L355-356: Please remove the last sentence since, in my opinion, speculations should be made in a Discussion section, and not in an Appendix.*

Done

*References*

*Hrachowitz, M., and M. P. Clark, 2017: HESS Opinions : The complementary merits of competing modelling philosophies in hydrology. Hydrol. Earth Syst. Sci., 21, 3953–3973, doi:10.5194/hess-21-3953-2017.*

---

## Author Response (AR3)

*Reviewers' comments are in italics.* Our responses are in blue.

Dear Editor,

We send you a revised version of our manuscript and a reply to the review comments we received.

We would like to stress that the first reviewer who had already reviewed the manuscript is satisfied with the revised version, and that the major modifications asked by the second reviewer are linked to the fact that he/she did not reviewed the revised version but the initial manuscript submission. Most of the modifications requested by this reviewer were already addressed in the past two review rounds. Therefore the new revised version includes only minor modifications, corresponding to the remaining suggestions not already addressed (see our reply for more detail).

We hope that this confusion will not lead you to ask for the article to be reviewed again by external reviewers, given the efforts we made to comply with the comments received in the three review rounds, and that you will be able to make a decision on the basis of our responses and the changes made in the article.

We thank you and the reviewers for your continued efforts to help us improving this manuscript.

Sincerely,

Guillaume Thirel

Reviewer #1

*The authors addressed my concerns. There are some small technical things that would be good to correct. There is a bracket missing in Table 1, it would be good to add to the caption of Figure 2 that this is a map of France and to explain the small land area in the bottom right. A final thorough look through the paper would be good to catch such things.*

We thank the reviewer for its assessment of the manuscript.

We added the bracket that was indeed missing. We also added in caption of Figure 2 that this is a map of France.

The small land area in the bottom right is the Corsica Island, which is part of France. We believe this is not necessary to mention it.

Reviewer #2

*The review of the manuscript "On the use of streamflow transformations for hydrological model calibrations" submitted to the Hydrology and Earth System Sciences.*

*The manuscript focuses on the analysis of the choice of the objective function and streamflow data transformation for the calibration and validation of the hydrological models. This study analysed three objective functions and 11 transformations using data from 1985-1995 and 1995-2005 from 325 catchments in France. The outcomes are presented for the specific single catchment, the Fecht at Wintzenhein, and averaged over 325 catchments.*

*The topic is very important and well-suited to the journal. I have some doubts regarding the applied methodology, the presentation of the results, and the quality of the discussion that should be improved.*

We thank the reviewer for his/her assessment of the manuscript. We must however say that we were at first rather surprised by some comments, as they did not seem to correspond to the content of the revised version of the manuscript we had sent to the editor after the previous review round. We then understood that the review comments were made on the initial version of the manuscript, which explains why they are not necessarily relevant anymore. We however tried to address them below, or to refer to our previous answers to reviews from Rounds #1 and #2 of reviews.

*General comments:*

(1) *The results are presented in the form of ranks; therefore, it is difficult to describe relative differences in the outcomes between tested transformations. The differences between methods may be not statistically significant. So, the ranking of the transformation method will be different when the other criteria are applied.*

We thank the reviewer for this legit comment. We do agree that the ranking of the transformations can and will probably be different when the methodology, including the criteria used to compare them, differs. Consequently, the conclusions we draw are impacted by our methodological choices, as are any results of any study. Regarding the use of ranks rather than another way of comparing transformations, we already addressed this issue (see the Answer to Comment #1 of Reviewer #2 of the Round #1 of review; https://egusphere.copernicus.org/preprints/2023/egusphere-2023-775/egusphere-2023-775-AC2-supplement.pdf). In the 2-page answer to this specific comment, we justified that the "idea behind choosing to work with ranks, instead of direct (normalised) errors was to i) be less impacted by different orders of errors magnitudes between catchments or ranges of streamflows, and to ii) answer the question of what are the best transformations, rather than how good transformations are." To answer the comment properly, we assessed the impact of modifying the methodology, i.e. using absolute differences rather than ranks. Based on results over the 325 catchments, we found "similar results as when working with ranks: the general conclusions are not changed. The same groups of transformations still are the best over the same ranges of streamflows.

As a consequence, we prefer not to change the whole methodology used in the manuscript and we will stick to the one proposed so far." We however modified the manuscript (see the first revised version) to mention this alternative and to discuss it.

(2) *The analysis covers two periods, 1985-1995, for the model's calibration and 1995-2005 and validation period. Are these periods overlapping? There is no description of hydroclimatic conditions during these periods. Was it dry, average or wet years? It is important as later data were classified into 200 intervals, and the ranks were evaluated for these classes.*

There is no overlapping between the periods. We indeed used hydrological years, i.e. from August to July. We added this information in the revised version of the manuscript.

We did provide information about the hydroclimatic conditions of both periods, through the calculation of mean air temperature, mean annual potential evapotranspiration, mean annual runoff, baseflow index, aridity index and aridity seasonality (see Table 2). As mentioned already in the

manuscript, this "shows that the climatic and hydrological conditions are similar between the two periods, with the evaluation period being only slightly warmer and wetter than the calibration period and other indicators showing only slight variations."

(3) *The quality of the calibration results is not addressed. What constitutes 'satisfactory performance'? Were optimal parameter values obtained for all catchments? How was calibration uncertainty considered? Were repeated calibrations conducted, and did they yield similar results (objective function values and ranks)?*

It is true that we did not provide much detail about the calibration results. The reason behind this is that the modelling platform and algorithms (i.e. the GR models, the airGR package and the optimization algorithm) we use have been used for years in many studies in many countries, including France of course, as they are developed there. The mentioned term "satisfactory results" was referring to that. We consequently added some references, two of them already cited in the manuscript, such as Perrin et al. (2003) who developed and assessed GR4J, Mathevet (2005) who compared this algorithm to SCE-UA among others, and Coron et al. (2017) who developed and assessed the airGR package.

As an additional indication allowing to state that calibration results were satisfactory, we calculated the average objective function among all models / objective functions / transformations / catchments, and we obtained a median value equal to 0.88 and an average value equal to 0.84. Please note that such scores only give as a qualitative assessment, since values coming from different objective functions and different transformations are hardly comparable. However, comparing these values to those of Table 3 of Crochemore et al. (2015), we can see that they can reasonably be considered as good or very good.

The question whether optimal parameter values were obtained for all catchments is difficult to answer. It is indeed impossible to answer that question without testing all combinations of parameter values, which is impossible to do due to the infinite number of possible combinations. Due to its deterministic nature, the optimization algorithm does not consider uncertainty and repeated calibrations necessarily lead to identical objective function values and parameters values and ranks. Assessing uncertainty in calibration certainly is an interesting topic, however we believe that it is beyond the scope of the current study.

(4) *Why was the Fecht at Wintzenhein chosen as an example catchment? Does it represent average conditions? Where is it located? Which hydrological model version was used?*

The Fecht River was chosen for illustrative purposes only. As a consequence, we only chose it because it could be used for showcasing the methodology. As we did not aim to use it to draw any conclusions (those are drawn from the results on the 325 catchments), we did not expect this catchment to represent any average conditions. The location of the Fecht catchment was added on Figure 2 in the previous rounds of review, as well as the name of the hydrological model (namely the GR4J model) in the caption of Figures 4 and 5.

*(5) The second part of the results presentation deals with the outcomes averaged over 325 catchments. How similar were the results across the 325 catchments? Were there distinct groups with different behaviours?*

Section 3.3 is already dedicated to an attempt to establish links between catchment characteristics and transformations. Unfortunately, as explained in this section, we failed to identify strong links between catchment characteristics and transformations, and consequently to identify groups with different behaviours.

*(6) The quality of the figures should be improved. In the case of Fig. 5, I recommend using a colour scale. Lines from Fig. 6, 9, 10, 11, and 12 have similar colours, so it is difficult to recognize what they are representing. The text regarding Figure 10 states that the transformation log and 0.2 show the best average ranks for the lowest flows. For me, there are four transformation methods that are characterized by almost the same values. Are these differences in the results significant?*

We thank the reviewer for the suggestion to improve the figures, which is consistent with comments we got in earlier reviews. The figure 5 from the initial submission was removed following reviewers' comments. Regarding initial figures 6, 9, 10, 11 and 12, following a reviewer's comment, we added symbols for the different transformations (see answer A20 in https://editor.copernicus.org/index.php?_mdl=msover_md&_jrl=778&_lcm=oc108lcm109w&_acm= get_comm_sup_file&_ms=110954&c=250088&salt=6749699781714668426). We believe these figures are much clearer now.

Regarding initial Figure 10 (now Figure 8), transformations log and 0.2 show a lower average rank over a large portion of flows (more than half of the range). Although some other transformations are not that far sometimes and although we did not compute statistical tests to assess the significance of the difference, we believe that the fact that this difference is visible over more than half of the flows range makes it worth mentioning.

*(7) I miss the discussion of the results, taking into account other studies. It is necessary to supplement the manuscript with a discussion of the results!*

We added a discussion of the results in the previous round of revision, in section 3.5 (see answer to major comment #2 of Reviewer #2 of the Round #2 of reviews).

*(8) What does it mean "Anti-correlations"?*

This expression, which was only present in the initial submission and was removed since, was used for meaning "negative correlation".

**Reference**:

Crochemore, L., Perrin, C., Andréassian, V., Ehret, U., Seibert, S. P., Grimaldi, S., … Paturel, J. E. (2015). Comparing expert judgement and numerical criteria for hydrograph evaluation. Hydrological Sciences Journal, 60(3), 402–423. https://doi.org/10.1080/02626667.2014.903331

Mathevet, T.: Quels modèles pluie-débit globaux au pas de temps horaire ? Développements empiriques et comparaison de modèles sur un large échantillon de bassins versants, Phd thesis, Doctorat spécialité Sciences de l'eau, ENGREF Paris, https://hal.inrae.fr/tel-02587642, 2005.